

# Greenland Monthly Accumulation Maps (1960-2022): A Statistical Semi-Empirical Bias-Adjustment Model

Josephine Lindsey-Clark[1], Aslak Grinsted[1], Baptiste Vandecrux[2], and Christine Schøtt Hvidberg[1]

[1]Physics of Ice, Climate, and Earth, Niels Bohr Institute, University of Copenhagen, Jagtvej 132, 2200 Copenhagen, Denmark
[2]Geological Survey of Denmark and Greenland, Øster Voldgade 10, 1350 Copenhagen, Denmark

**Correspondence:** Josephine Lindsey-Clark (josephine.lindsey-clark@nbi.ku.dk)

**Abstract.**

Accurate estimates of snow accumulation over the Greenland Ice Sheet (GrIS) are essential for reliable projections of sea-level rise. These are typically obtained from Regional Climate Models (RCMs), which carry substantial temporal and spatially variable biases, contributing to the metre-scale uncertainties in sea-level rise projections. While numerous studies have eval-

uated RCM bias using select in-situ observational datasets, many assessments are deduced from comparison to reanalysis datasets, which too carry substantial uncertainties. Such biases stem partly from the inability of RCMs and reanalysis products to assimilate point-based in-situ precipitation measurements directly. As a result, the rich network of observations from radar, ice cores, snow pits and stake networks remains under-utilised in systematic bias-correction of model accumulation.

In this study, we present a novel statistical-semi-empirical model for bias-correcting gridded accumulation output from any

RCM or reanalysis product, utilising two million observational data points from the SUMup surface mass balance dataset. The method applies an empirical orthogonal function (EOF) decomposition to model accumulation output and adjusts the mean, climatology, EOFs and corresponding principle components (PCs) through a set of coefficients. The coefficients are calibrated by using a least squares optimisation that minimises the misfit between each component of the model accumulation and the in-situ observations. This allows us to reconstruct spatially complete bias-corrected accumulation maps. Here we apply this

method to monthly accumulation output from HIRHAM5 (1960-2022), RACMO 2.4p1 (1980-2022), and CARRA reanalysis (1991-2022), identifying initial mean biases of -8.7% (HIRHAM), +0.5% (RACMO) and +10.9% (CARRA). After adjustment, these are reduced to -0.1%, -0.1% and -0.2%, respectively. Resulting bias-corrected mean annual accumulation rates over the ice sheet are estimated at 321 mm yr$^{-1}$ (HIRHAM, 1960-2022), 375 mm yr$^{-1}$ (RACMO, 1980-2022) and 384 mm yr$^{-1}$ (CARRA, 1991-2022).

The framework outlined in this study offers a scalable, transferable solution for enhancing accumulation estimates, applicable to other climate models, variables, regions and observational datasets. The resulting bias-corrected accumulation fields offer an improved input to ice-sheet models, with the potential to reduce uncertainties in future sea-level rise projections through enhanced integration of observational data.



# 1 Introduction

The Greenland Ice Sheet (GrIS) has become the greatest single contributor to present-day global sea-level rise (Hofer et al., 2020; Fettweis et al., 2020; van den Broeke et al., 2016), accounting for approximately 22% of the $\sim$3.3 mm yr$^{-1}$ total mean sea-level rise between 2002 to 2022 (Hanna et al., 2024; Jia et al., 2022). Since the mid 1990s, mass loss from GrIS has been driven by changes in Surface Mass Balance (SMB) (Hofer et al., 2020; van den Broeke et al., 2016, 2009), overtaking ice loss from calving. SMB is defined as the mass accumulated through precipitation, minus the mass lost through meltwater runoff,

sublimation, evaporation, and wind redistribution. As the largest component of SMB, accumulation is crucial to constrain for accurate modelling of ice-sheet evolution. However, due to the high spatial and temporal complexity of precipitation patterns, accumulation over the GrIS remains poorly constrained. As a result, regional climate models (RCMs) often fail to adequately capture this variability, leading to biased estimations of ice mass loss (Hanna et al., 2024) and substantial discrepancies between climate model projections (Otosaka et al., 2023).

Understanding and quantifying accumulation over the GrIS has been a long-standing challenge in polar research. The first studies investigating snow accumulation began in the early 20th century, establishing techniques such as stake measurements, snow pits, and shallow cores to deduce accumulation from snow height and seasonal layering. Ice core projects starting in the late 1950s analysing stable oxygen-isotope ratios provided new insights into past climate, later leading to the systematic use of ice cores for detailed accumulation studies (e.g. Box et al., 2013, 2009; Buchardt et al., 2012; Mosley-Thompson

et al., 2001; Clausen et al., 1988). In the mid-1960s, the introduction of ice-penetrating radar systems to track reflections from internal ice layers revealed that stratified ice layers formed by seasonal snowfall could be used to infer accumulation rates. Unlike ice cores, snow pits and stake measurements–which provided point observations–radar techniques enabled continuous mapping over vast areas. Early radar studies, such as (Robin et al., 1969), demonstrated the effectiveness of radar in revealing accumulation variability over broad regions, laying the foundation for further ground-based radar surveys (Hawley et al., 2014;

Miège et al., 2013; Medley et al., 2013), as well as airborne campaigns (Montgomery et al., 2020; Lewis et al., 2017; Koenig et al., 2016). These advances allowed for systematic coverage of regions of the ice sheet that were previously inaccessible to in situ measurements, dramatically improving our understanding of accumulation patterns and spatial variability.

One of the earliest comprehensive efforts to synthesise a diverse range of observational data to map accumulation across GrIS was by Ohmura and Reeh (1991). Using spatial interpolation of measurements from snow pits, ice cores and coastal

weather stations, they estimated the mean accumulation over GrIS to be 310 mm.w.e, and provided one of the most accurate maps of accumulation over the ice sheet at the time. This influential study played a crucial role in improving understanding of how topography and weather systems influence regional snowfall patterns, and the impact on the ice sheet's mass balance. Subsequent studies incorporated additional data with improved interpolation techniques (Cogley, 2004; Calanca et al., 2000; Ohmura et al., 1999), and enhanced understanding of regional variability (McConnell et al., 2001).

Regional climate models provided a way to estimate accumulation patterns on a spatially complete, high resolution grid (Ettema et al., 2009; Fettweis et al., 2008; Box et al., 2006; Box, 2005; Box et al., 2004). Similarly, climate reanalysis data were used to provide accumulation grids (Hanna et al., 2008, 2006, 2005), with the advantage of assimilating in situ atmo-



spheric observations such as pressure and temperature. Reanalysis datasets could also serve as boundary conditions for RCM simulations, alongside ancillary observational datasets such as those from weather stations and remote sensing. However, both RCMs and reanalysis products inherently carry biases due to limitations in model physics and sparse observational constraints.

To address model bias, Box et al. (2006) calibrated accumulation output from the Fifth Generation Mesoscale Model for polar climates (Polar MM5) using snow pit observations, identifying and correcting for systematic errors. Expanding on these advances, Burgess et al. (2010) combined firn core measurements and meteorological station precipitation data with high-resolution Polar MM5 output to create a spatially complete reconstruction of Greenland Ice Sheet accumulation. This hybrid methodology resolved inconsistencies in earlier studies that relied on sparse datasets or models alone, providing a new accumulation grid with enhanced regional accuracy. Using spatial interpolation of linear correction functions derived by region, Burgess found a mean snow accumulation rate of $337 \pm 48$ mm yr$^{-1}$ w.e, 16-21% higher than previous estimates by Ohmura et al. (1999), Calanca et al. (2000) and Cogley (2004). This increase was primarily attributed to better representation of higher orographic precipitation, affecting the south east in particular–a region with limited ice core coverage. Accumulation rates in the south-east were found to exceed 2000 mm yr$^{-1}$ and dominate the inter-annual variability. Representing 31% of the total accumulated mass, this region was found to have a substantial impact on the ice-sheet surface mass balance as a whole, highlighting the importance of studying regional variability.

Providing one of the first spatially complete reconstructions of accumulation, Burgess et al. (2010) remains a cornerstone for understanding Greenland's climate-driven ice loss. Since then, numerous studies have continued to improve on this foundation through incorporating new observational data and enhanced model simulations (Mouginot et al., 2019; Sandberg Sørensen et al., 2018; van den Broeke et al., 2016; Khan et al., 2015; Velicogna et al., 2014; Box et al., 2013; Shepherd et al., 2012). Despite these advances, SMB remains a major source of uncertainty in projections of future sea-level rise (van den Broeke et al., 2009).

Ice sheet models, and thus sea-level rise projections, require spatially complete gridded accumulation maps and are therefore typically obtained from RCMs. RCMs are often validated using a combination of remote sensing data and in-situ point observations from weather stations and, occasionally, firn cores. However, as remote sensing technology cannot accurately measure surface mass fluxes such as snowfall (Bennartz et al., 2019) and the sparse distribution of weather stations and firn cores leaving vast regions without data coverage, RCM accumulation maps still carry significant uncertainties today (Vernon et al., 2013). Though in situ accumulation data from ice cores, radar, snow pits and stake measurements can fill some of these in-situ data gaps, challenges in aligning the inconsistent temporal resolutions means that the full range of available data remains under-utilised in systematic RCM validation.

Here we present a flexible statistical-semi-empirical model designed to utilise the full range of available in-situ data to bias-adjust any gridded model accumulation output. The SUMup dataset (Vandecrux et al., 2024) includes a compilation of SMB data derived from a diverse range of different sources including radar, ice-core, snow pit and stake measurements, providing an extensive basis for model correction. Using this set of over two million data points in Greenland, we produce a data cube of monthly bias-adjusted spatially complete accumulation maps. While previous studies have used techniques such as universal kriging (Ohmura and Reeh, 1991), triangulated irregular networks (Burgess et al., 2010) and least-squares regressions (Box



et al., 2013) to estimate or correct model fields with point-based observations, these approaches typically do not explicitly account for patterns of both spatial and temporal bias variability.

We base our method on Empirical Orthogonal Function (EOF) decomposition of the model accumulation output; first centring the data to remove the temporal mean and climatology, and then computing the first 10 EOFs and corresponding Principle Components (PCs) on the anomaly matrix. The EOF modes reveal the preferred patterns of spatial variability captured in the model accumulation, while the principle components describe how each mode contributes to the variability through time. Utilising robust least-squares optimisation, we fit the model data to the SUMup dataset to derive a set of coefficients which adjust

each component of the decomposition. This enables targeted correction of the model's spatial and temporal structure, offering physical interpretability of how each component influences model bias. Limiting the adjustment to the first 10 EOF modes captures 90% of the variability, while avoiding over-fitting noise captured by the higher order modes.

The aim of this paper is to present the method alongside key data considerations, as well as providing bias-corrected accumulation maps from three models: the HIRHAM regional climate model version 5, the Polar Regional Atmospheric Climate

Model (RACMO) version 2.4p1, and the Copernicus Arctic Regional Reanalysis (CARRA) west domain. We focus on the accumulation zone, a region characterised by negligible runoff, making precipitation the most dominant component of the accumulation. While the method could be extended to adjust full SMB over the whole ice sheet with additional data, our focus here is to test the approach using key variables available across models. This ensures a consistent comparison, targeting the largest component of SMB, rather than multiple, less well constrained physical elements simultaneously. We provide an anal-

ysis of mean and seasonal biases before and after bias-adjustment and examine the impact on long-term accumulation trends and their sensitivity to temperature.

## 2   Data

### 2.1   Gridded Model Accumulation

We present bias-corrected accumulation maps from three models including two RCMs and one reanalysis dataset: (1) the

HIRHAM regional climate model version 5, forced with ERA5 reanalysis (Hersbach et al., 2020) between January 1960 to December 2022, (2) the Regional Atmospheric Climate Model (RACMO) version 2.4p1 also forced with ERA5 reanalysis, between January 1980 and December 2022, and (3) the Copernicus Arctic Regional Reanalysis (CARRA) data between January 1991 and December 2022. For each model, accumulation is obtained by merging the total precipitation (rainfall and snowfall) with the evaporation and sublimation fields, providing an appropriate representation of net accumulation within the accumu-

lation zone. The specific variable names for each model are stated below. All output data is obtained at monthly temporal resolution.





### 2.1.1 HIRHAM5 Regional Climate Model (1960-2022)

HIRHAM5 (DMI et al., 2017), is the fifth version of the HIRHAM regional atmospheric climate model, which combines the dynamics of the HIRLAM model (Undén et al., 2002) with the physical parametrisation schemes of the ECHAM model (Roeckner et al., 2003). The model is run on a rotated latitude–longitude grid with 5.5 km horizontal resolution over the Greenland/Iceland domain. The precipitation and evaporation/sublimation fields are the variables named 'pr', which includes rainfall and snowfall, and 'evspsbl'–evaporation including sublimation and transpiration.

### 2.1.2 RACMO 2.4p1 Regional Climate Model (1980-2022)

RACMO 2.4p1 (van Dalum et al., 2024), is a hydrostatic model that integrates the atmospheric dynamics of HIRLAM (Undén et al., 2002) version 5.0.3 with the physical parametrisations of the ECMWF Integrated Forecasting System. Here, the variables 'pr' (rainfall + snowfall), and 'evspsbl' (evaporation including sublimation and transpiration) are used from the R24 experiment (van Dalum et al., 2024) on the pan-Arctic domain grid on a 11 km grid.

### 2.1.3 CARRA Reanalysis (1991-2022)

The CARRA reanalysis dataset (Copernicus Climate Change Service, 2021a) is produced using the HARMONIE-AROME non-hydrostatic regional numerical weather prediction model on a 2.5 km grid, providing the highest resolution gridded model data available over the Arctic. CARRA reanalysis uses ERA5 global reanalysis as lateral boundary conditions for input together with additional observational data. The west domain has a 1069x1269 lambert conformal grid covering Greenland. The variable 'tp', total precipitation, is merged with the variables 'eva', evaporation, and 'tisef', time integrated snow evaporation flux, which are the closest equivalent variables to 'evspsbl' used for HIRHAM and RACMO. Monthly data is obtained by downloading the 30 hour and 6 hour forecasts, which are subtracted to obtain daily values and then resampled to monthly sums.

## 2.2 Analysis Domain

We apply a mask to restrict the analysis domain to the interior accumulation zone. This zone is defined as the area where both HIRHAM and RACMO simulate no runoff; the CARRA runoff product is excluded from this criterion due to known quality issues affecting its reliability. This masked region designates where bias adjustment is performed, and only SUMup observations falling within this area are used in the analysis and discussion. We focus on accumulation rather than net surface mass balance for several reasons: (1) observations within the accumulation zone are less impacted by melt-related uncertainties and thus are generally more reliable; (2) we gauge that the SUMup dataset does not provide sufficient information on melt processes to reliably constrain runoff biases; and (3) extrapolating biases derived from interior observations of SMB to lower-elevation melt zones would likely be unreliable due to differing processes and conditions.

Inside the accumulation zone mask, where simulated runoff is zero, accumulation, $Acc$, is approximated as $Acc = P - E$, where $P$ is precipitation and $E$ is evaporation/sublimation. This approximation neglects smaller components such as melt, re-freezing, wind redistribution and drifting snow erosion, which are not represented in precipitation and evaporation/sublimation



| Variable | Bias [mm yr$^{-1}$] | Bias [%] | R | RMSE |
|----------|------------|----------|------|--------|
| RACMO P-E | +1.6 | +0.5 | 0.91 | 0.0068 |
| RACMO SMB | -5.3 | -1.7 | 0.91 | 0.0067 |

**Table 1.** Model mean bias, correlation ($R$), and root mean square error (RMSE) before bias-adjusting with SUMup data. Bias is expressed both in mm yr$^{-1}$ and as the mean percentage deviation from in-situ observations.

variables. While SUMup observations significantly affected by melt are not included in the dataset, the cumulative influence of these un-modelled processes may contribute to model-observation discrepancies in some regions within the mask, potentially affecting the reliability of the approximation.

In RACMO2.4p1, a full SMB variable is available, which includes variables for drifting snow erosion, $D$, and sublimation including wind blown snow, $S$. RACMO SMB is defined as $SMB = P - R - D - S$ (van Dalum et al., 2024). We evaluate the performance of RACMO $P - E$ versus $SMB$ by a point-wise comparison with SUMup data inside the mask (Table 1). We find slightly reduced RMSE for $SMB$ (0.0067) versus $P - E$ (0.0068) with near identical correlation (0.91), while absolute bias and percentage bias are smaller for $P - E$ (+1.6 mm yr$^{-1}$, +0.5%) compared to $SMB$ (-5.3 mm yr$^{-1}$, -1.7%). The greater bias for $SMB$ may point to uncertainties in the additional drifting snow erosion and wind blown sublimation terms, which are difficult to validate directly, or reflect measurement uncertainties in the SUMup data. Given the comparable performance, we asses that $P - E$ is sufficient for our analysis, as uncertainties in both model and observational values likely overshadow any differences observed. This choice facilitates a consistent comparison with HIRHAM, which does not include drifting snow erosion and wind-blown snow sublimation in SMB (Langen et al., 2017), and CARRA, for which a dedicated SMB product is not available.

## 2.3 SUMup In-Situ Observational Data

The SUMup collaborative database is a compilation of in situ measurements of SMB as well as subsurface temperature and density for the Greenland and Antarctic ice sheet from published and unpublished sources (Vandecrux et al., 2024). The 2024 edition contains, in an harmonised format, more than 2.4 million SMB values given between two dates or, if dates are unknown, between two years. Of these, over 1.7 million lie within the accumulation zone, derived primarily from airborne radar (Montgomery et al., 2020; Lewis et al., 2017; Koenig et al., 2016), ground based radar (Lewis et al., 2019; Miège et al., 2013), ice/firn-cores (Kawakami et al., 2023; Freitag et al., 2022a, b, c; Vinther et al., 2022; Osman et al., 2021; Lewis et al., 2019; Graeter et al., 2018; Miège et al., 2013; Box et al., 2013; Hanna et al., 2011; Bales et al., 2009; Banta and McConnell, 2007; Hanna et al., 2006; Mosley-Thompson et al., 2001; Miller and Schwager, 2000d, a, b, c, 2004; Hammer and Dahl-Jensen, 1999; Clausen et al., 1988), snow pits (Kjær et al., 2021; Schaller et al., 2016; Bolzan and Strobel, 2001a, b, 1999a, b, c, d, e, f, g, 1994) and stake measurements (Dibb and Fahnestock, 2004).

Most accumulation estimates in SUMup, particularly those derived from radar and ice cores, rely on density corrections to convert observed layer thicknesses into snow-water-equivalent values. This introduces additional uncertainty, especially where



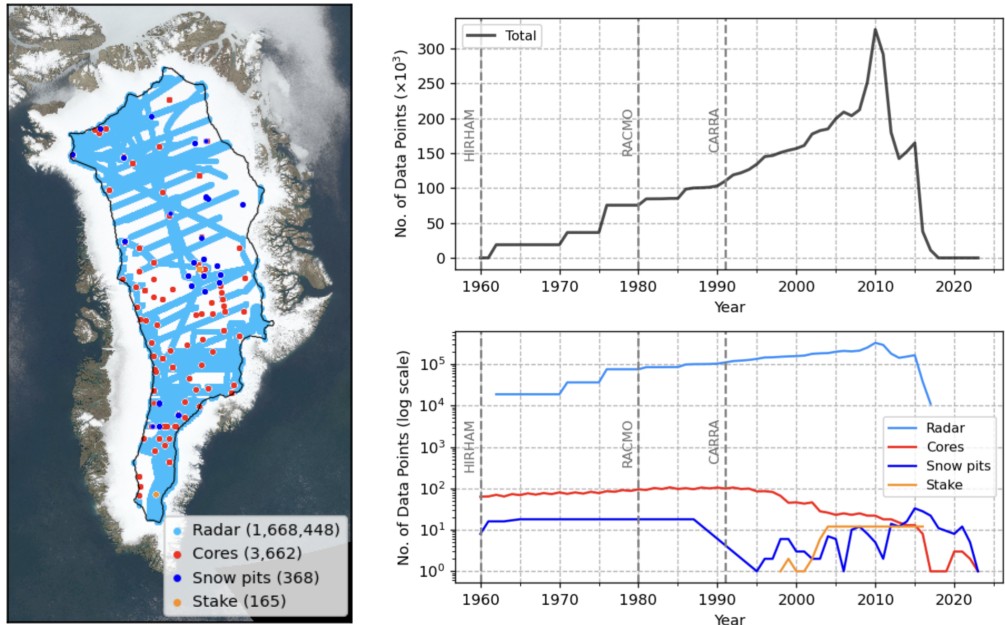

**Figure 1.** Left: geographical distribution of SUMup data points used in this study between 1960 and 2022. The colour of map markers indicate the method used to derive the accumulation data, with the total number of data points available for each method type in this time period. Right upper: time dependence of total number of data points with markers for the start year of model data. Right lower: time dependence of data points by measurement method, with counts in log scale.

density profiles are sparse, inferred, or inconsistently applied. Prior to bias-adjustment, the SUMup data is filtered to remove suspected errors, including negative radar-derived accumulation, which likely reflect processing or calculation errors. After detailed inspection of data from Koenig et al. (2016), we identified that accumulation estimates derived from the 2009 and 2010 surveys appeared largely unrealistic, containing many values either less than half or more than double the corresponding model estimates and also contradicting other measurements in the SUMup dataset. Koenig et al. (2016) also reported worse statistics for 2009 and 2010 than for other years, when comparing radar-derived values to accumulation from the MAR regional climate model. Consequently, we assume that values from the 2009 and 2010 surveys are subject to improper horizon identification or density assumptions and are excluded from the analysis. Additionally, Koenig et al. (2016) state that the picked end-of-melt-season horizons could be assigned an approximate date of 1st July. Based on multiple examples of melt or rain events in late July (Tedesco and Fettweis, 2020), August (Box et al., 2022) and some time as late as October (Harper et al., 2023), the date of these horizons in the 2024 SUMup release was set to 1st September. Lastly, we found discrepancies between the timestamps associated to accumulation values in Miège et al. (2013) and the related dataset Miege et al. (2014). After testing several possibilities, pre-summer start dates were set to 1st January, while post-summer start dates are set to 1st January the following year. Similarly, pre-summer end date are changed to 31st December of the previous year and post-summer end date are changed





to 31st December. This interpretation, based on typical seasonal snowfall patterns, was seen to improve the consistency with
195 the rest of the dataset and model estimates.

Fig. 1 shows the geographical distribution of SUMup data points used in this study and their time dependence. A breakdown
of their spatial coverage by decade is provided in Fig. A2 in the Appendix.

The temporal resolution of the in-situ data varies between method types, where ice core and radar measurements are primarily
based on annual and multi-annual periods. Snow pit measurements are provided at varying resolution between 8-12 months,
while stake measurements cover between 1-9 months. As shown in table 2, the vast majority of data points are annual or
multi-annual means, making up 84-88%.

For measurements where only the start year and end year are provided in the SUMup dataset, new start date and end date
fields are introduced based on the original studies if stated. For example, some sources measuring cores with $H_2O_2$ dating use
the distance between summer peaks to derive annual accumulation (e.g. Kjær et al., 2021), while others use the mid-winter peak
(e.g. Miège et al., 2013). When start and end dates are not explicitly provided in the source text, default values corresponding
to the first and last days of the reported year(s) are assigned. This applies to 37% of entries for HIRHAM (1960–2022), 28%
for RACMO (1980–2022), and 18% for CARRA (1991–2022). Dates are rounded to the nearest month start/end to limit the
temporal precision to monthly resolution to match the temporal precision of the RCM output.

We assign a dating uncertainty to every SUMup record to account for errors in the time bounds of each SMB measurement,
stated in months. This uncertainty is taken from the original study, if stated, otherwise, a default value is assigned based on
commonly stated dating uncertainties in studies using similar methods. For radar and ice/firn-core measurements, we assign a
dating uncertainty of 12 months, to account for the possibility of a missed annual layer. Snow pits, for which precise start dates
are unknown, are assigned a dating uncertainty of 2 months, while stake measurements which have defined start and end dates
are given a dating uncertainty of 0. The dating uncertainty is assigned to each measurement as a gaussian weighted uncertainty
distribution, allowing the measurements to be bias-adjusted with the monthly RCM data within the given uncertainty range.

The latitude and longitude coordinates of each measurement are then matched with the nearest grid cell from the respective
model. To reduce computational time required for the fitting, radar data points from the same source and year range–which are
thus not independent–are grouped and averaged within grid cells. This leads to a different number of individual data points for
fitting each model due to their different spatial resolutions. As illustrated in table 2, for the same 1991-2022 period, HIRHAM
has 69,643 points for the fitting, while RACMO with lower resolution has 30,062, and CARRA has 150,862–over twice as
many HIRHAM and over five times as many as RACMO. This approach effectively smooths radar data to a greater degree for
the lower-resolution models.

Fig. 2 shows a point-wise comparison of each SUMup data point (with radar values averaged within grid cells) matched
with the corresponding model grid point value. With respect to the SUMup data, HIRHAM shows a negative percentage bias
and standard deviation of the three models, underestimating mean annual accumulation by -8.3% with a standard deviation of
85.3 mm yr$^{-1}$ for the 1991-2022 overlap period. RACMO exhibits the lowest mean bias of 1.3%, and a standard deviation
of 81.0 mm yr$^{-1}$ for the same period, which is lower for the 1980-2022 period at 0.5%. CARRA shows the greatest positive
bias of +10.9%, and the lowest standard deviation of 79.6 mm yr$^{-1}$. These differences likely reflect a combination of model




| Period | Model | Annual/multi-annual | Sub-annual | Monthly | Total |
|--------|-------|---------------------|------------|---------|-------|
| 1960–2022 | HIRHAM | 92,743 (88.0%) | 12,487 (11.8%) | 161 (0.2%) | 105,391 |
| 1980–2022 | HIRHAM | 77,770 (86.0%) | 12,487 (13.8%) | 161 (0.2%) | 90,418 |
| | RACMO | 34,139 (85.8%) | 5,466 (13.7%) | 161 (0.4%) | 39,766 |
| 1991–2022 | HIRHAM | 69,643 (84.6%) | 12,478 (15.2%) | 161 (0.2%) | 82,282 |
| | RACMO | 30,062 (84.3%) | 5,457 (15.3%) | 161 (0.5%) | 35,680 |
| | CARRA | 150,862 (85.1%) | 26,200 (14.8%) | 161 (0.1%) | 177,223 |

**Table 2.** Number and percentage of observational data points fitted for each model and time period by type and total.

resolution, parametrisation schemes, as well as potential issues with observational data, which are discussed in greater detail
in section 4. A small number of points with anomalously high negative bias—visible as secondary and tertiary peaks in the
log-scale histogram (Fig. 2)—originate from airborne radar estimates by Koenig et al. (2016) and Montgomery et al. (2020).
These may arise from an incorrect density conversion or missed annual layers in radargrams, resulting in two or more years
of accumulation being recorded as one. Apart from the data from the Koenig et al. (2016) 2009 and 2010 surveys that we
discarded, we could not find specific criteria to isolate these potentially erroneous measurements, and since the radar-derived
SMB observations bring a highly valuable spatial coverage, we retain the remaining radar measurements.

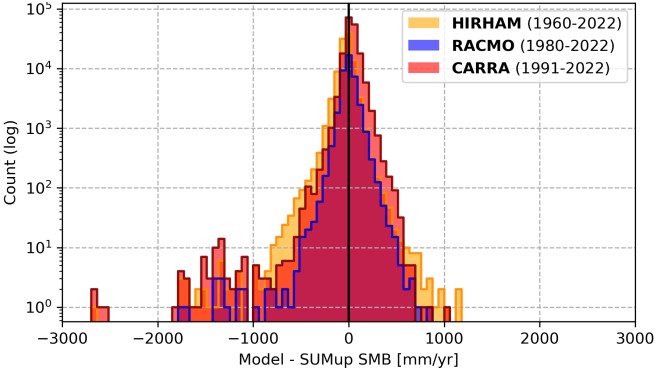

**Figure 2.** Left: histogram of model bias before bias-correction (model - SUMup data) covering the full time period of available data for each
model. Right: table showing the mean bias for different time periods.

# 3 Methods

## 3.1 EOF analysis

The EOFs are obtained by performing a Principle Component Analysis (PCA) on the masked RCM accumulation output, using
the eofs.xarray module from the eofs Python library (Dawson, 2016).





The model data, $\mathbf{X}(x,y,t)$, is first centred by removing the temporal mean, $\mathbf{M}(x,y)$, and the climatology, $\mathbf{C}(x,y,m)$, where $\mathbf{C}$ is the mean deviation of each month, $m$, from the temporal mean:

$$\mathbf{C}(x,y,m) = \frac{12}{N_t} \sum_{i=1}^{N_t/12} (\mathbf{X}(x,y,12(i-1)+m) - \mathbf{M}(x,y)), \tag{1}$$

where $m = \mathrm{month}(t)$ and $N_t$ is the total number of months. The centred data, $\mathbf{X}'(x,y,t)$ is thereby expressed as:

$$\mathbf{X}'(x,y,t) = \mathbf{X}(x,y,t) - \mathbf{M}(x,y) - \mathbf{C}(x,y,m). \tag{2}$$

The centred data, $\mathbf{X}'(x,y,t)$, is masked to the accumulation zone and rearranged into matrix form $\mathbf{X}'$, where each row ($i$) corresponds to a time index, and each column, ($j$), represents a location, $(x_j,y_j)$, within the mask. As the regional model grids are optimised for near-equal area, no area weighting is applied during the analysis.

To compute the EOFs, the covariance matrix, $\mathbf{\Psi}$, of the anomalies, $\mathbf{X}'$, is calculated as:

$$\mathbf{\Psi} = \frac{1}{N_t - 1} \mathbf{X}'^{\mathrm{T}} \mathbf{X}'. \tag{3}$$

The EOFs are derived by solving the eigenvalue problem, $\mathbf{\Psi}\mathbf{E} = \mathbf{E}\mathbf{\Lambda}$, where the eigenvectors, $\mathbf{E}$, are the EOFs and $\mathbf{\Lambda}$ are the eigenvalues arranged in a diagonal matrix. The EOFs are ordered by their corresponding eigenvalues such that the first EOF captures the greatest amount of variance in the data. The Principle Components (PCs) are obtained by projecting the centred data onto the EOFs:

$$\mathbf{PC} = \mathbf{E}^T \mathbf{X}', \tag{4}$$

where each column of $\mathbf{PC}$ represents the time series of the corresponding mode, expressing how much each mode contributes to the variability through time. The PCs are computed at the temporal resolution of the RCM, which is monthly in this analysis.

The reconstructed RCM accumulation, $\mathbf{RCM}(x,y,t)$ can thereby be expressed as:

$$\mathbf{RCM}(x,y,t) = \mathbf{M}(x,y,t) + \mathbf{C}(x,y,m) + \sum_{i=1}^{N} \mathbf{PC}_i(t) \cdot \mathbf{EOF}_i(x,y). \tag{5}$$

In this study, the first 10 EOF modes are used, capturing over 90% of the variance in each of the models. Using a truncation such as this, a truncation residual term, $\mathbf{R}(x,y,t)$, is introduced to describe the remaining variability and noise not captured by the first 10 modes.

It is common to de-trend data prior to EOF decomposition to prevent a long-term linear or low-frequency signal from dominating the leading modes. However, as accumulation trends vary strongly by region and no clear trend was seen to dominate the PCs, there was no advantage in removing a single linear trend over the accumulation zone as a whole. By not de-trending the data, the resulting modes can reflect both the variability and long-term evolution, allowing the bias-adjustment to account for trends in a way that appropriately scales with the rest of the data.

To extrapolate values beyond the accumulation zone mask, EOF maps for the full ice-sheet, $\mathbf{EOF}_{\mathrm{full}}$, are calculated using the PCs derived from the EOF decomposition of the accumulation zone. For the full ice sheet, the anomalies, $\mathbf{X}'_{full}(x,y,t)$ can





be expressed as:

$$\mathbf{X}'_{full}(x,y,t) = \sum_{i=1}^{N_t} \mathbf{PC}_i(t) \cdot \mathbf{EOF}_{full,i}(x,y). \tag{6}$$

Eq. 6 is projected onto $\mathbf{PC}_i(t)$ and rearranged to obtain an expression for $\mathbf{EOF}_{full,i}(x,y)$:

$$\mathbf{EOF}_{\text{full},i}(x,y) = \frac{\mathbf{PC}_i(t) \cdot \mathbf{X}'_{full}(x,y,t)}{\sum\limits_{i=1}^{N_t} \mathbf{PC}_i^2}. \tag{7}$$

## 3.2 The model

The SUMup dataset is used to find a set of coefficients to adjust each component of the reconstructed model accumulation, derived by fitting the in-situ SUMup data points using the python package scipy.optimize.least_squares for least-squares optimisation (Virtanen et al., 2020).

We introduce a set of parameters to adjust the mean and EOFs (temporally independent) and climatology and PCs (temporally dependent) components of the reconstruction. The bias-adjusted reconstructed accumulation, $\mathbf{Acc}(x,y,t)$ is thereby expressed as:

$$\mathbf{Acc}(x,y,t) = a_0 + \mathbf{M}(x,y,t) + b_0\mathbf{C}(x,y,m) + \sum_{i=1}^{10}(b_i\mathbf{PC}_i(t) + a_i)\,\mathbf{EOF}_i(x,y) + \mathbf{R}(x,y,t), \tag{8}$$

where $a_0$ adjusts the temporal mean, $\mathbf{M}(x,y)$, $a_i$ adjust the EOFs, $\mathbf{EOF}_i$, $b_0$ adjusts the climatology, $\mathbf{C}$, $b_i$ adjusts the PCs, $\mathbf{PC}_i$, and the truncation residual term, $\mathbf{R}$ is not adjusted. The initial guess (no adjustment) is set to 0 for the $a$ parameters and 1 for the $b$ parameters. Using 10 EOFS, there are 22 unknowns.

We minimise the misfit between the SUMup observations and the modelled grid-point values using a residual formulation that accounts for the varying temporal coverage of the observational data. To ensure that each data point contributes appropriately to the overall fit, we scale the residual by dividing by the number of months spanned by the observation and multiplying by twelve, normalising to an annual scale. This effectively gives greater weight to higher resolution data containing more temporal detail.

Due to the temporal uncertainty inherent in much of the SUMup dataset, of which over 84% consists of annual or multi-annual averages, the precise timing of individual accumulation events is often uncertain in the observations. Therefore, to prevent over-fitting and mitigate the impact of dating errors, we implement Tikhonov regularisation on the $b$-parameters, constraining them towards the initial guess via a regularization parameter $\lambda$. The optimum $\lambda$ is found through 5-fold cross-validation, determined by where the average mean squared error (MSE) of the residuals is minimised. To ensure that the cross-validation process accounts for potential dependencies within SUMup, we ensure that temporally or spatially linked measurements—such as data points from the same ice core, snow pit, stake network, or radar horizon—are grouped and assigned to the same fold during cross-validation. The resulting optimal $\lambda$ values were 12.6 for HIRHAM, 8.9 for RACMO, and 7.9 for CARRA, reflecting differences in how each model's bias structure affects the fit, as determined independently by the cross-validation procedure.





Several data transformation techniques, including log and arcsinh transforms, were tested to reduce the influence of outliers.
The impact of each transformation was evaluated using the same 5-fold cross-validation approach described above. However,
these provided no measurable improvement in fit quality, and thus development and optimisation of the method continued in
linear space. As an alternative, we evaluated the five loss functions available in the scipy.optimize.least_squares package–linear,
soft_l1, huber, cauchy and arctan–across a range of outlier threshold values (f_scale). We found that the arctan loss function
with f_scale = 1, yielded the greatest reduction in MSE for HIRHAM and RACMO, while still achieving an acceptable im-
provement for CARRA. Based on the residual distributions shown in Fig. 2, we infer a high probability that observations
differing by more than 1 myr$^{-1}$ from all three models can be regarded as outliers. We therefore adopt the arctan loss function
with f_scale = 1 in our final configuration.

### 3.3 Bias Analysis

The coefficients derived from fitting the output data from each model are used to reconstruct the bias-adjusted accumulation ac-
cording to eq. 8. As the models are bias-adjusted with SUMup data only within the accumulation zone mask, the reconstructed
data for the full ice-sheet is extrapolated using eq. 7. The model bias is presented both as the absolute difference (original mean
minus bias-adjusted mean), as well as the percentage difference, calculated at each grid point as the following:

$$Bias\ \% = \frac{\overline{X} - \overline{X'}}{\overline{X}} \cdot 100, \tag{9}$$

where $\overline{X}$ is the original model temporal mean and $\overline{X'}$ is the bias-adjusted temporal mean. The seasonal bias, grouped by winter
(DJF), spring (MAM), summer (JJA) and autumn (SON), is also assessed by the same metrics.

The effect of the bias-adjustment is also assessed through analysis of the linear trends and the accumulation sensitivity to
temperature, both analysed by grid-point-wise and domain-wide regressions. The sensitivity is defined as the percent change
in accumulation per degree Celsius increase in temperature, derived from the Clausius-Clapeyron relationship (Clausius, 1850;
Clapeyron, 2006) describing the saturation water vapour pressure, $e_s$ as a function of temperature, $T$. Using a similar approach
to Nicola et al. (2023) and Bochow et al. (2024), we can express the Clausius-Clapeyron relationship as:

$$\frac{\mathrm{d}\ln e_s}{\mathrm{d}T} = \frac{L}{R_v T^2} = k, \tag{10}$$

where $L$ is the latent heat of vaporisation, $R_v$ is the specific gas constant for water vapour and $k$ is the growth constant.
Assuming precipitation, $P$, scales with the saturation vapour pressure, it is common to model the response of precipitation
to temperature via a log-linear relationship (Nicola et al., 2023; Bochow et al., 2024). In this study, we extend this approach
to net accumulation, here defined as $(\text{Acc} = P - E)$, where $E$ represents evaporation/sublimation. We assume that under
cold and/or high-latitude conditions where sublimation and evaporation are limited relative to precipitation, accumulation
approximately scales with precipitation. Therefore we fit $\ln(Acc)$ against Northern Hemisphere mean temperature anomalies
from the Hadley Centre dataset (Morice et al., 2021, HadCRUT5 analysis), which combines near-surface (2 m) air temperature
and sea surface temperature (SST) anomalies. The growth factor, $k$, is determined from the slope of the linear regression, and
therefore represents the fractional change in precipitation per degree Celsius. To express this as a percent change, we define





the sensitivity, $s$, expressed by Bochow et al. (2024):

$$s = 100 \times \left(e^k - 1\right),\qquad(11)$$

where $s$ is the percent change in accumulation per degree of warming.

## 4 Results

### 4.1 Bias-Adjustment Parameters

The bias-adjustment coefficients derived from fitting each of the three model outputs to SUMup SMB data within the accumulation zone mask are displayed in Fig. 3. Coefficients are shown for the full available model period, as well as the shorter periods of overlap. The 1991-2022 period covered by all 3 models is highlighted for comparison. Error bars reflect confidence in the fitted parameters rather than spatial variability.

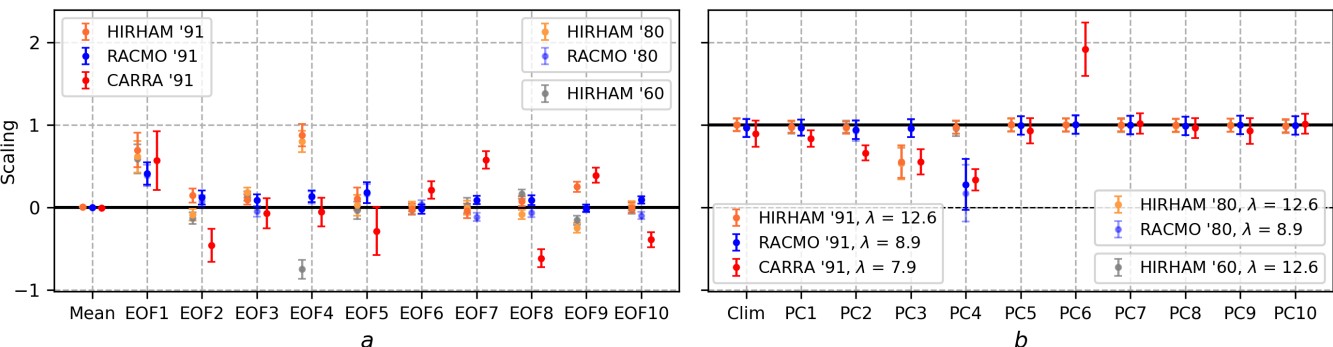

**Figure 3.** Bias-adjustment coefficients derived from fitting HIRHAM, RACMO and CARRA data using the arctan loss function, with error bars reflecting uncertainty in the parameter estimates. The coefficients derived from each model period are highlighted separately for comparison. The initial guess (no adjustment) is denoted by the solid black line. Left: the time-independent $a$ parameters adjusting the mean bias and 10 EOF modes. Right: the $b$ coefficients adjusting the time-dependent climatology and PC modes, with $\lambda$ value derived from cross-validation.

The $a$ parameters adjust spatial patterns with no temporal dependency. The coefficient adjusting the mean bias is close to 0 for all models over all periods, ranging from a minimum of -0.003 for CARRA and a maximum of 0.007 for HIRHAM between 1991-2022. This suggests that all models capture the mean accumulation relatively well relative to observations, with minimal systematic bias.

The EOF coefficients adjust the amplitude of the spatial patterns of variability. The corresponding EOF maps for the first 345    three modes in each model can be found in Appendix Fig. A3. Among the model EOF coefficients, CARRA shows the greatest variation and widest error bars, with a mean deviation from the initial guess of 0.36 and a standard error of 0.41. For the same 1991-2022 period, HIRHAM and RACMO show smaller mean deviations of 0.23 and 0.12, respectively, and smaller uncertainties with standard errors of 0.3 and 0.11.



The $b$ coefficients adjust the time-dependent components of the accumulation reconstruction in eq. 5. Their values are influ-
enced by the $\lambda$ regularisation parameter, applied to penalise deviations from the initial guess due to the temporal uncertainty in
dating the observational data and the limited number of monthly measurements (table 2). Lambda values are derived for each
model individually through cross-validation. For HIRHAM, the highest lambda value of 12.6 limits deviation from the initial
guess more aggressively than for RACMO and CARRA with values of 8.9 and 7.9, respectively. This is reflected by a mean
deviation from the initial guess of 0.06 for HIRHAM, 0.1 for RACMO and 0.28 for CARRA, between 1991-2022.

For all three models, the coefficient adjusting the climatology is within $\pm 0.1$ of the initial guess, leaving the seasonal signal
largely unchanged. Few PC adjustment coefficients ($b_1 - b_{10}$) deviate substantially from their initial value of 1 for all three
models. The most notable exceptions are HIRHAM PC3, RACMO PC4, and CARRA PCs 4 and 6, which are adjusted by 0.5,
0.2, 0.3, and 2 respectively.

### 4.2 Bias-Adjustment Assessment Metrics

Table 3 presents the model mean bias, correlation and RMSE before and after bias-adjustment based on point by point compar-
isons–i.e., each SUMup data point (with radar values averaged within grid cells) matched with the corresponding model grid
point value.

Prior to bias-adjustment, CARRA exhibits the largest mean bias of +10.9%. This is followed by HIRHAM, showing negative
mean bias of -8.3% to -8.8%, while RACMO exhibits a minimal initial bias of +0.5% for 1980-2022 and +1.3% for 1991-
2022. After adjustment, the mean bias is reduced to near-zero in all models; -0.1% for HIRHAM and RACMO, and -0.2%
for CARRA, with substantial reductions in the high initial bias for HIRHAM and CARRA. Notably, all models, including
RACMO, show increased correlation after bias-adjustment, improving by 1-3%. In addition, RMSE is reduced for all models
by 8-18%, with the smallest reductions for RACMO and the largest for CARRA. A comparison of model performance against
observations from 6 individual ice core sites before and after bias-adjustment is provided in Fig. A5.

| Period | Model | Bias [%] | Bias$_{adj}$ [%] | R | R$_{adj}$ | $\Delta$R [%] | RMSE | RMSE$_{adj}$ | $\Delta$RMSE [%] |
|---|---|---|---|---|---|---|---|---|---|
| 1960–2022 | HIRHAM | -8.7 | -0.1 | 0.87 | 0.90 | 2.4 | 0.0074 | 0.0065 | -12.2 |
| 1980–2022 | HIRHAM | -8.8 | -0.1 | 0.88 | 0.91 | 2.8 | 0.0074 | 0.0063 | -14.4 |
| | RACMO | +0.5 | -0.1 | 0.91 | 0.92 | 0.9 | 0.0068 | 0.0062 | -8.1 |
| 1991–2022 | HIRHAM | -8.3 | -0.1 | 0.88 | 0.91 | 3.0 | 0.0074 | 0.0063 | -15.3 |
| | RACMO | +1.3 | -0.1 | 0.91 | 0.92 | 1.0 | 0.0068 | 0.0062 | -8.5 |
| | CARRA | +10.9 | -0.2 | 0.90 | 0.92 | 1.8 | 0.0071 | 0.0058 | -18.0 |

**Table 3.** Model mean bias, correlation ($R$), and root mean square error (RMSE) before and after bias-adjusting with SUMup data. Bias is
expressed as the mean percentage deviation from in-situ observations.





### 4.3 Mean Accumulation and Spatial Bias Patterns

Fig. 4 shows spatially complete mean fields for original model values, bias-adjusted values and mean bias, expressed as the absolute difference as well as the percentage bias, calculated by eq. 9. Spatial patterns are shown here for HIRHAM (1960-2022), RACMO (1980-2022) and CARRA (1991-2022), and periods of overlap are provided in the appendix, Fig. A1. Mean values for the full periods as well as overlaps are presented in Table 4 (in mm yr$^{-1}$) and Table A1 in the Appendix (in Gt yr$^{-1}$). As the models are bias-adjusted with SUMup data only within the accumulation zone mask, detailed analysis of spatial pattern biases will be focussed on the region inside the accumulation zone mask.

Before bias-adjustment, the three models share a similar overall spatial pattern: low accumulation in the north-east, gradually increasing southward, with the highest values in the south and south-east, increasing rapidly towards the ice sheet margins. After bias-adjustment, this overall behaviour remains largely unchanged, however the bias patterns reveal considerable differences between models. While HIRHAM underestimates accumulation across the majority of the ice-sheet, CARRA overestimates almost everywhere. RACMO's bias pattern shares features with both HIRHAM and CARRA, resembling HIRHAM across the northern two-thirds of the ice-sheet but appearing more similar to CARRA in the southern third. One consistent feature across all three models is a region of positive bias in the north west, where accumulation is overestimated increasingly towards the margins. This shared tendency is also seen to a lesser degree at the central west and central east margins outside of the accumulation zone.

HIRHAM is consistently biased low over all time periods, underestimating by -12.4% in the accumulation zone and -8.4% over the entire ice sheet for the full 1960-2022 period. This bias diminishes as earlier data is excluded (4). Prior to bias-adjustment, a distinct area of low accumulation (0-100 mm yr$^{-1}$) is apparent in the northern part of the accumulation zone, which is no longer evident after bias-adjustment. This change is reflected in the map of percentage bias, which shows that the areas surrounding the ice divide in the north-east carry the most negative bias, with values reaching up to -60%. The map of absolute bias reveals that the highest accumulation regions in the south-east also contribute significantly to model error, where HIRHAM underestimates accumulation by over 120 mm yr$^{-1}$ towards the ice sheet margins.

RACMO shows the smallest average bias of the three models, underestimating accumulation by -4.9% in the accumulation zone and -1.3% over the ice sheet for the full 1980-2022 period. Before bias-adjustment, RACMO also shows a distinct area of low accumulation (0-100 mm yr$^{-1}$) in the northern interior, which vanishes after bias-adjustment. The strongest negative percentage bias in RACMO is likewise located in the north-east interior, reaching up to -40%. However, towards the south, a prominent positive absolute bias emerges in RACMO, where values exceed +150 mm yr$^{-1}$ towards the south west ice-sheet margins. In contrast, a small area of negative bias is observed at the south-east margin, reaching up to -60 mm yr$^{-1}$.

CARRA shows the highest accumulation of all three models, overestimating by a mean of 8.6% in the accumulation zone and 9.6% over the ice sheet. Before bias-adjustment, the area of low accumulation in the northern interior–visible in both HIRHAM and RACMO–is not present in CARRA, which shows that the mean accumulation does not fall below 100 mm yr$^{-1}$ anywhere within the accumulation zone. However, a small region with values of 0-100 mm yr$^{-1}$ does emerge after bias-adjustment. Like




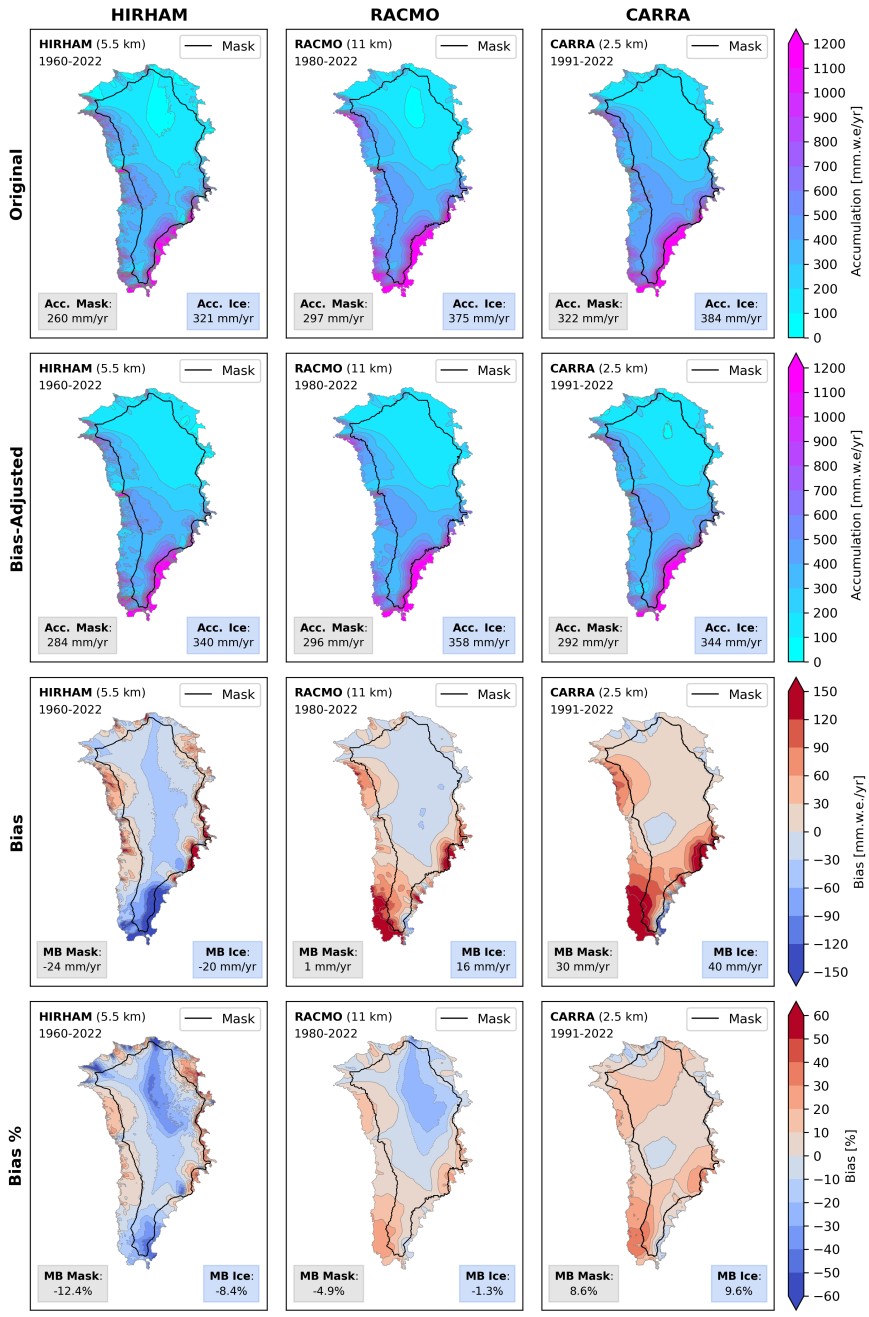

**Figure 4.** Mean maps of model accumulation and bias over for HIRHAM (1960-2022), RACMO (1980-2022) and CARRA (1991-2022). Values stated in the lower corners of each plot indicate the spatial mean calculated over the accumulation zone (left) and ice sheet (right). First row: original mean model accumulation before bias-adjustment. Second row: mean bias-corrected accumulation. Third row: mean bias in mm yr$^{-1}$, (original - bias-adjusted). Fourth row: mean percentage bias, mean(original - bias-adjusted) / mean(original).



| Period | Model | Accumulation Zone | | | | Ice Sheet | | | |
|---|---|---|---|---|---|---|---|---|---|
| | | Acc | Acc$_{adj}$ | Bias | Bias % | Acc | Acc$_{adj}$ | Bias | Bias % |
| | | [mm yr$^{-1}$] | [mm yr$^{-1}$] | [mm yr$^{-1}$] | [%] | [mm yr$^{-1}$] | [mm yr$^{-1}$] | [mm yr$^{-1}$] | [%] |
| 1960–2022 | HIRHAM | 260 | 284 | −24 | −12.4% | 321 | 340 | −20 | −8.4% |
| 1980–2022 | HIRHAM | 265 | 289 | −24 | −11.7% | 326 | 346 | −20 | −7.9% |
| | RACMO | 297 | 296 | +1 | −4.9% | 375 | 358 | +16 | −1.3% |
| 1991–2022 | HIRHAM | 268 | 290 | −22 | −10.4% | 329 | 348 | −18 | −6.9% |
| | RACMO | 301 | 298 | +3 | −3.5% | 379 | 361 | +18 | −0.1% |
| | CARRA | 322 | 292 | +30 | +8.6% | 384 | 344 | +40 | +9.6% |

**Table 4.** Mean annual accumulation in the accumulation zone and ice sheet before and after bias adjustment. Bias is shown in both absolute (mm yr$^{-1}$) and relative percentage (%). Corresponding values in Gt yr$^{-1}$ are provided in Table A1 in the Appendix.

RACMO, CARRA exhibits its strongest positive bias in the south-west, where values exceed +150 mm yr$^{-1}$, contrasted by a narrow region of negative bias in the south east.

### 4.4 Seasonal Bias Maps

Fig. 5 shows the mean seasonal percentage bias for each model over the full model period. It is important to note that 84–88% of the SUMup observations used in this study are annual or multi-annual means, with many start and end dates estimated from mid-winter or mid-summer measurement peaks. Among the sub-annual records, most span 8–10 months, where the end date is when the measurement was taken–typically between April and July–and the start date is estimated as the previous summer peak, often assumed to be 1st July (e.g. Montgomery et al., 2020; Lewis et al., 2017) or September (assigned in SUMup for snowpits from Vandecrux et al. (2023). Only a small fraction (0.1-0.5%) are monthly measurements matching the temporal resolution of the model output. Thus, the seasonal bias patterns should be interpreted within the context of these limitations, as they are unlikely to be as robust as the long-term annual means. For this reason, only the full model periods are examined without comparisons across the overlap periods.

For HIRHAM, the highest mean negative bias occurs during the winter and spring months, reaching up to -16% in the accumulation zone and -11% over the ice sheet. This bias is reduced in the summer, decreasing to -10.5% and -6.5%, before increasing slightly in the autumn to 10.6% and 7.3%, respectively. The overall spatial pattern remains relatively consistent through the year, with regions of strongest negative bias around the ice divide in the northern interior and in the south. In the winter, the northern interior shows a large region where bias exceeds -50%. This area is reduced in size in the spring, reaching a maximum of -40% in small regions in the summer.

RACMO also shows the lowest mean bias in the winter and spring months in the accumulation zone, at -7%, becoming less negative for the summer and autumn months at -4%. However for the whole ice sheet, which includes larger areas of positive bias, the winter and spring mean biases are only -2%, while the summer shows the greatest negative bias at -5%, becoming



**Figure 5.** Model mean seasonal percentage bias: mean(original - bias-adjusted)/mean(original). Mean seasonal patterns are shown for HIRHAM between 1960-2022, RACMO between 1980-2022 and CARRA between 1991-2022.





least negative in autumn at -1%. Though overall spatial patterns are relatively consistent throughout the year, maximum biases
in the south-west reach up to +40% in the winter and gradually decrease through the year to +20% in autumn. Negative biases
in the north east are also more pronounced in the winter and spring reaching -50%, reducing to up to -40% in summer and
autumn.

CARRA shows a consistent overall positive bias throughout the year. In the accumulation zone, the highest mean bias occurs
during the autumn and summer months at +9%, reducing to +8% in the spring and winter months. Over the full ice sheet, the
lowest bias is also seen in spring and winter at +9%, with the highest bias in the summer at +12%. In the winter, regions of
negative bias are seen in the north and central west interior, with values in the accumulation zone between 0 and -20%. These
areas reduce in size through the year, while other areas of negative bias emerge in the north-east. In the autumn, almost no areas
of negative bias remain within the accumulation zone, leaving a relatively uniform and low positive bias of less than +30%.

## 4.5 Temporal Trends

Fig. 6 shows the temporal trends captured in the full model data before and after bias-adjustment. The maps represent the
spatial variation in trend fits, calculated by regressing each grid-point accumulation over the full data period. Below, the mean
annual accumulation over the whole domain (accumulation zone or ice sheet) is plotted against time with the linear fit. Mean
accumulation trends resulting from fitting the models to each of the three time periods are shown in table 5.

HIRHAM shows a positive trend across the accumulation zone between 1960-2022. The rate of accumulation increase is
highest towards the south and central east margins, reaching over 30 mm yr$^{-1}$ decade$^{-1}$. There are few small areas which
show a decrease in accumulation, in the central west, north and south, all of which are outside the accumulation zone mask.
This pattern remains largely unchanged after bias-adjustment, and is reflected by the linear trends for both domains decreasing
by only 0.1 mm yr$^{-1}$ decade$^{-1}$.

For RACMO between 1980-2022, areas of decreasing accumulation rates are observed in the south west and along the north
margin before bias-adjustment. In addition, a small pocket of sharply decreasing accumulation rates is present above areas of
strong positive increasing accumulation in the south east. After bias-adjustment, the negative area in the south west are now
positive, while the negative area in the north now covers a wider region, stretching further south into the accumulation zone.
The adjustment decreases mean trends over the accumulation zone and ice sheet more significantly than for HIRHAM, from
5.7 mm yr$^{-1}$ decade$^{-1}$ and 7.4 mm yr$^{-1}$ decade$^{-1}$, to 4.7 mm yr$^{-1}$ decade$^{-1}$ and 6.8 mm yr$^{-1}$ decade$^{-1}$ respectively.

CARRA (1991-2022) shows negative trends spanning most of the western ice sheet, with the exception of small areas near
the north-west margins and in the south-central-west, while the north and south east are dominated by positive trends. After
bias-adjustment, the negative areas are interrupted by a region of positive trends covering the central ice sheet from west to
east. More positive trends appear across the centre, while larger areas of more intense negative trends exceeding -30 mm yr$^{-1}$
decade$^{-1}$ emerge in the north west. CARRA shows the lowest mean trends of the three models of 1.0 mm yr$^{-1}$ decade$^{-1}$ in
the accumulation zone and 1.8 mm yr$^{-1}$ decade$^{-1}$ over the ice sheet, reducing to 0.5 mm yr$^{-1}$ decade$^{-1}$ and 1.5 mm yr$^{-1}$
decade$^{-1}$ after bias-adjustment.



**Figure 6.** Temporal regressions before and after bias-adjustment shown for HIRHAM (1960-2022), RACMO (1980-2022) and CARRA (1991-2022). Upper: maps showing the grid-point-wise regression representing spatial variation in linear trends. Lower: mean annual accumulation over the accumulation zone (mask) and full ice sheet with linear fits.

Mean accumulation trends resulting from fitting the models to each of the three time periods are shown in 5. Accumulation trends decrease unanimously after bias-adjustment across all models and domains. For the bias-adjusted HIRHAM and



RACMO data, it can be seen that trends decrease as earlier data is excluded, while uncertainties and p-values increase. The
temporal trend in bias for each model is further analysed in Fig. A4, illustrating how each model bias evolves through time.

| Period | Model | Accumulation Zone Trend [mm yr$^{-1}$ decade$^{-1}$] | Accumulation Zone Trend$_{adj}$ [mm yr$^{-1}$ decade$^{-1}$] | Ice Sheet Trend [mm yr$^{-1}$ decade$^{-1}$] | Ice Sheet Trend$_{adj}$ [mm yr$^{-1}$ decade$^{-1}$] |
|---|---|---|---|---|---|
| 1960–2022 | HIRHAM | $4.6 \pm 3.5$, p=0.01 | $4.5 \pm 3.4$, p=0.01 | $5.1 \pm 4.4$, p=0.02 | $5.0 \pm 4.3$, p=0.03 |
| 1980–2022 | HIRHAM | $4.4 \pm 6.4$, p=0.18 | $4.1 \pm 6.3$, p=0.20 | $5.2 \pm 8.2$, p=0.21 | $4.9 \pm 8.0$, p=0.23 |
|  | RACMO | $5.7 \pm 7.1$, p=0.12 | $4.7 \pm 7.0$, p=0.18 | $7.4 \pm 8.9$, p=0.10 | $6.8 \pm 8.7$, p=0.13 |
| 1991–2022 | HIRHAM | $3.1 \pm 10.0$, p=0.54 | $3.0 \pm 9.9$, p=0.55 | $3.3 \pm 13.2$, p=0.62 | $3.2 \pm 12.9$, p=0.62 |
|  | RACMO | $4.0 \pm 11.4$, p=0.49 | $3.2 \pm 11.1$, p=0.57 | $5.6 \pm 14.6$, p=0.45 | $5.1 \pm 14.2$, p=0.48 |
|  | CARRA | $1.0 \pm 12.2$, p=0.88 | $0.5 \pm 9.7$, p=0.92 | $1.8 \pm 14.8$, p=0.81 | $1.5 \pm 11.9$, p=0.80 |

**Table 5.** Mean temporal trends calculated over the full time series and overlap periods, with confidence intervals and p-values.

## 4.6 Temperature Sensitivity Analysis

Fig. 7 shows the results of the temperature sensitivity analysis, calculated according to eq. 10. The maps represent the spatial
variation in sensitivity, $s$, defined in eq. 10, resulting from the grid-point wise regression of log-scaled model accumulation
against northern hemisphere temperature (NHT) anomalies. White regions inside the ice sheet area are caused by negative
accumulation values, which are undefined. Below, the domain-wide mean annual accumulation is plotted in log-scale against
NHT anomalies with the resulting linear fit and sensitivities indicated in the legend.

HIRHAM (1960-2022) exhibits the highest mean sensitivities of all three models. Before bias-adjustment, these are 8.1%K$^{-1}$
for the accumulation zone and 7.1%K$^{-1}$ over the ice-sheet, with areas across the central- to north-east exceeding 25%K$^{-1}$ to-
wards the ice-sheet margins. Negative sensitivities reaching up to 15%K$^{-1}$ are observed in narrow regions near the north-,
central- and south-western margins outside the accumulation zone, though positive sensitivities dominate the ice sheet over-
all. After bias-adjustment, mean sensitivities are reduced by 8-12% to values of 7.1%K$^{-1}$ (accumulation zone) and 6.8%K$^{-1}$
(ice-sheet). While the overall spatial pattern remains largely the same, the regions of highest sensitivity (>10%K$^{-1}$) in the
north-east are slightly reduced in size and shift further east. This leaves the sensitivity across the west and south largely be-
tween 0-5%K$^{-1}$.

RACMO (1980-2022) shows lower mean sensitivities with values of 6.8%K$^{-1}$ (accumulation zone) and 5.7%K$^{-1}$ (ice-
sheet) before bias-adjustment. Like HIRHAM, RACMO also exhibits areas of high sensitivity across the central- to north-east
reaching up to a maximum of 25%K$^{-1}$, though these are slightly smaller in size and are located further inland. Areas of negative
sensitivity of between 0-20%K$^{-1}$ are present in the north-west towards the ice sheet margins, appearing more prominently
than in HIRHAM. In the south, RACMO shows high positive sensitivities in the east reaching up to 20%K$^{-1}$, contrasted with
negative sensitivities in the west up to -5%K$^{-1}$. After bias-adjustment, mean sensitivities are reduced by 27-35%, to values



**Figure 7.** Temperature sensitivity analysis for HIRHAM (1960-2022), RACMO (1980-2022) and CARRA (1991-2022). Upper: maps representing spatial patterns of sensitivity derived from grid-point wise regressions of log-scaled model accumulation against NHT anomalies. Empty grid points resulting from negative values in the accumulation are excluded in the calculation of the mean sensitivities. Lower: accumulation zone wide and ice-sheet wide mean annual accumulation (log-scaled) plotted against NHT anomalies with linear regressions and sensitivities indicated in the legend.





of 4.7%K$^{-1}$ in the accumulation zone and 4.1%K$^{-1}$ over the ice-sheet. The highest positive values in the east are reduced in magnitude, now reaching up to a maximum of 20%K$^{-1}$. Regions of negative values in the north now reach further inland, while the region of negative sensitivity in the south-west is reduced in size.

CARRA (1991-2022) shows the lowest mean sensitivities of all three models, with values of 1.9%K$^{-1}$ over the accumulation zone and 0.7%K$^{-1}$ over the ice sheet before bias-adjustment. Negative sensitivities dominate most of the west side of the accumulation zone, contrasted with a region of high sensitivities reaching up to a maximum of 25%K$^{-1}$ in the central- to north-east–a feature common to all three models. A west-east contrast is observed in the south, with negative sensitivities reaching up to -25%K$^{-1}$ in the south-west and +25%K$^{-1}$ in the south-east. After bias-adjustment, the magnitude of both positive and negative sensitivities is enhanced, creating stronger spatial contrasts. Mean sensitivities are reduced to 1.2%K$^{-1}$
(accumulation zone) and 0.3%K$^{-1}$ (ice-sheet), decreasing by 35-55%. The region of highest positive values in the north- to central-east expands, with a larger proportion exceeding 25%K$^{-1}$. The negative region in the north also expands further inland, with a large areas of values between -15%K$^{-1}$ to -25%K$^{-1}$.

Sensitivities derived from the domain-wide regressions are lower in most cases. An exception is RACMO over the ice sheet, where the domain-wide sensitivity is higher at 5.5%K$^{-1}$ compared to the point-wise mean of 4.2%K$^{-1}$, and notably, CARRA
exhibits negative domain-wide sensitivities for the accumulation zone.

## 5 Discussion

### 5.1 Mean Bias and Seasonal Bias Patterns

For all models, the bias-correction substantially improves mean percentage bias, correlation and RMSE compared to observations (3). Inter-model agreement also improves significantly, with the standard deviation between models decreasing from
27 mm yr$^{-1}$ to 4 mm yr$^{-1}$ after adjustment. Interestingly, a consistent slightly negative bias of -0.1% to -0.2% remains in all models, despite initial positive biases in RACMO and CARRA. This demonstrates that the robust loss function successfully limits the influence of outliers, which are mostly negative (Fig. 2). As the RMSE metric gives full weight to the outliers, it thus becomes negative.

Despite having the coarsest resolution of the three models, RACMO, shows the lowest initial point-wise mean bias of +0.5%
for the full 1980-2022 period and 1.3% for 1991-2022. Comparison of the initial accumulation fields with the bias corrected fields also shows that RACMO has the lowest mean biases, ranging from -0.1% to -4.9% (Fig. 4). Inspection of the spatial bias patterns, however, reveals that RACMO exhibits areas of both strong positive and negative bias. These spatial contrasts may explain its low overall mean bias, while CARRA and HIRHAM, with significantly higher mean biases, have more consistent positive and negative spatial bias patterns, respectively.

CARRA exhibits the highest initial point-wise mean bias of 10.9%, as well as high mean biases in the spatially complete fields of 8.6% in the accumulation zone and 9.6% for the whole ice-sheet. With the highest resolution (2.5 km), one would expect CARRA's accumulation estimates to be the most accurate the three models. Yet, we find it to overestimates accumulation across the majority of ice-sheet, with bias exceeding +150 mm yr$^{-1}$ in the south west. Previous studies have found that CARRA



outperforms other models compared to observations. Comparing CARRA and RACMO2.3p2 with daily snow depth data from
9 coastal weather stations, van der Schot et al. (2024) report that correlation coefficients between CARRA and observations are
generally higher than for RACMO, with no clear overestimation or underestimation from either model. Similarly, (Box et al.,
2023) evaluate CARRA, ERA5, NHM-SMAP, RACMO and MAR against in-situ precipitation data from 7 sites in southern
Greenland, finding that CARRA simulations correlate highest with field data from the relatively wet sites. They note, however,
that correlations at drier locations vary substantially due to limited data resulting in less robust statistics. These studies are
based on small, mostly coastal datasets, and do not reflect conditions across the broader ice sheet. In contrast, our analysis of
CARRA uses over 170,000 observational data points spanning the entire accumulation zone over 31 years, revealing biases
that may not be captured in more localised evaluations.

The CARRA Data User Guide (Copernicus Climate Change Service, 2021b) notes that the model tends to overestimates
precipitation compared to in-situ observations over northern Norway, Sweden and Finland. It also highlights, that the model
is in better agreement with the observations of high-precipitation events than ERA5, whose coarser resolution is not able to
predict the highest precipitation amounts. Hence, CARRA's tendency towards overestimation may in fact be linked to its higher
spatial resolution, which enhances its sensitivity to intense precipitation events, potentially amplifying them beyond what is
observed.

Common to all models is a strong bias in the south; a region characterised by high snowfall and mountainous topography.
Previous studies have identified the south-east, in particular, as a significant source of uncertainty, due to high spatial variability,
complex orographic effects and limited observational coverage (Burgess et al., 2010; Miège et al., 2013; Koenig et al., 2016).
This south-eastern bias is most prominent in HIRHAM. Langen et al. (2015) identify that while HIRHAM tends to overestimate
precipitation near the coastal margin due to overly strong topographic enhancement, lee-side precipitation is underestimated,
consistent with (Herrera et al., 2010). With the coastal margin excluded from our accumulation-zone-only analysis, the strong
negative bias observed here may reflect this lee-side underestimation.

While HIRHAM underestimates across the southern ice sheet, RACMO and CARRA show only a narrow region of negative
bias along the south-eastern margin, and substantial overestimation in the south-west. These contrasting bias patterns may
stem from differences in how each model represents topography–particularly important in southern Greenland, where steep
gradients and strong orographic effects make precipitation highly sensitive to resolution and parametrisation schemes.

The northern interior of the ice-sheet also contributes significantly to model bias in HIRHAM and RACMO. This may reveal
a shared tendency between these models to underestimate accumulation at cold, dry, high elevation sites, where atmospheric
moisture is limited and snowfall events are infrequent, but significant for SMB. Langen et al. (2017) also report that at high-
elevation sites, HIRHAM5-based simulations underestimate net accumulation by 8–16%.

It is important to consider the spatial and temporal coverage of observations used in the bias adjustment, which is illustrated
by decade in Fig. A2. Although the ice-sheet interior has relatively good and consistent coverage through time, the south-east
lacks data prior to 2000. Additionally, the south-west remains poorly observed throughout the entire period, with only a few
radar profiles and three core sites. As a result, adjustments in the south-east are less well constrained in the early record, while
those in the south-west rely on limited input throughout, contributing to greater uncertainty in both regions. Nonetheless, the





south-east has frequently been identified as a significant source of bias in earlier studies (Burgess et al., 2010; Miège et al., 2013; Langen et al., 2015; Koenig et al., 2016), supporting the need for substantial correction in this region.

In table 4, mean biases for HIRHAM and RACMO are seen to decrease as earlier data is excluded, suggesting that model performance improves in more recent periods. This trend may be influenced by two factors. Firstly, SMB observations become increasingly uncertain further back in time: deeper ice-core layers are more prone to miscounting, and radar horizons are more likely to be misidentified. As a result, bias adjustments based on these older data may introduce more, possibly unnecessary, corrections. Secondly, the number of observations assimilated into ERA5, which provides boundary conditions for all three models, declines drastically in earlier decades, degrading model performance in the past.

The maps of seasonal bias (Fig. 5) reveal that bias patterns are broadly consistent throughout the year in all three models. Mean seasonal bias values, however, fluctuate slightly between seasons, to a greater degree for HIRHAM and RACMO. Greater winter and spring biases over the accumulation zone in these models may be influenced by stronger winds during the winter and spring months. As the model accumulation is here represented only by the precipitation and evaporation/sublimation fields, wind-driven processes most important in the windier winter and spring months are not accounted for, possibly contributing the higher mean bias values in these months. This seasonal shift may also reflect limitations in how the models represent precipitation processes and orographic effects with respect to storm systems, which vary considerably over the course of the year. During winter and spring, when synoptic-scale storms are more frequent and spatially complex, moisture-rich air is more likely to be lifted over steep terrain. Coarser model grids (e.g., in HIRHAM and RACMO) may struggle to fully capture the resulting higher snowfall, contributing to the stronger negative biases observed during these seasons, particularly affecting regions with complex terrain. Conversely, CARRA, with higher spatial resolution, shows less pronounced seasonal variability in bias, which may indicate a better representation of these processes—though its overall positive bias also suggests possible overcorrection. However, as discussed in section 4.4, such speculations about the sources of seasonal biases should be viewed within the context of the temporal limitations of the SUMup data.

## 5.2 Temporal Trends and Temperature Sensitivity

A notable outcome of the bias-correction is the consistent reduction in accumulation trends across all models and time periods following adjustment. This suggests that the bias-adjustment tempers model-derived accumulation trends. In addition, trends are seen to decrease as earlier data is excluded (table 5). This may reflect a combination of stronger bias corrections applied to earlier, less well-constrained model data and a higher density of observations in recent decades.

Among the models, CARRA exhibits the lowest and least significant trends, paired with wide confidence intervals and p-values of 0.8-0.9, indicating a high degree of uncertainty. Estimates derived from shorter time series are inherently more uncertain due to both the natural variability and reduced statistical confidence. In contrast, HIRHAM's longer historical coverage (1960–2022) provides more statistically robust estimates, supported by lower uncertainties and statistically significant p-values of 0.01 within the accumulation zone and 0.02-0.03 over the ice-sheet. These longer-term trends are more resilient to inter-annual variability, which can dominate shorter records and obscure underlying trends. As such, the longer HIRHAM-based estimates, spanning 1960–2022, likely provide the most reliable assessment of long-term accumulation trends.





The temperature sensitivity estimates are similarly less robust over shorter time-scales (Fig. 7). An important distinction arises when comparing the point-wise mean sensitivities to those derived from domain wide regressions below. In most

cases, domain-wide sensitivities are lower, reflecting the effect of spatial averaging before performing the regression. Notably, CARRA exhibits negative domain-wide sensitivities for the accumulation zone—further reflecting the instability of trends over this shorter period. In contrast, mean sensitivities for HIRHAM and RACMO show greater consistency between the two regression approaches, supporting the robustness of their long-term sensitivity estimates.

The spatial pattern in the RACMO derived temperature sensitivities resembles that found by Buchardt et al. (2012), who

derive sensitivities using ice cores at 52 locations across the Greenland ice-sheet. They find the highest sensitivities over the central ice-sheet of $9.2\pm1.0\%\text{K}^{-1}$ (central east) and $9.4\pm0.1\%\text{K}^{-1}$ (central west), and the lowest sensitivities in the north-east $(1.5\pm2.8\%\text{K}^{-1})$ and north-west $(6.7\pm0.2\%\text{K}^{-1})$. RACMO also shows a band of higher sensitivities across the centre, with values decreasing further north. Similarly, in the south, where RACMO shows positive sensitivities in the east contrasted with negative values in the west, Buchardt et al. (2012) find a sensitivity of $8.2\pm0.8\%\text{K}^{-1}$ in the east and $-4.0\pm0.1\%\text{K}^{-1}$ in the

west.

### 5.3 EOF Analysis and Interpretation of Adjustment Coefficients

Interpretation of the EOFs and their adjustment coefficients (Fig. 3) may offer additional insights into model bias structure. The minimal adjustment of the mean bias coefficient for all three models suggests that a uniform offset is not the dominant source of error, and that biases are more effectively corrected through adjustments to the EOF and PC components. Similarly, the

climatology parameter remains within $\pm0.1$ of the initial guess after adjustment. As the majority of the fitted data is made up of annual or multi-annual means, short-term or highly seasonal fluctuations are unlikely to be represented in the observational data, making the limited adjustment of the climatology both reasonable and expected.

The coefficients adjusting the EOFs representing spatial patterns of variability differ significantly between models. The larger variation and higher uncertainty in CARRA may point to greater sensitivity to localised processes or to internal variability as a

result of the higher resolution grid. In addition, the smaller grid cell sizes have implications for the comparison between model output and observations: for radar-derived accumulation measurements which are grouped by source and averaged within grid cells, fewer measurement points are grouped within each grid cell. This leads to a greater number of points to fit for CARRA – over twice as many than HIRHAM and almost five times as many for RACMO for the same time period (table 2). For the other independent measurement types (cores, snow pits and stake measurements) which are not grouped, the finer grid makes it less

likely that multiple observations fall within a single cell. These factors increase the degrees of freedom in the fit, but also the potential for over-fitting and noise amplification, particularly in areas with sparse or uneven coverage. With fewer observations per cell, local discrepancies carry more weight, leading to greater variability in the fitted coefficients. Conversely, the coarser resolutions of HIRHAM and RACMO average more observational points within each grid cell, effectively smoothing localised anomalies and providing a more stable fit during optimisation. This is reflected by smaller mean deviations in their coefficients

from the initial guess and lower uncertainties.



The PCs coefficient adjustments may give insight into systematic temporal biases and potential links to large-scale climate drivers. Previous studies have linked leading EOF modes in Greenland snowfall with the North Atlantic Oscillation (NAO; Wong et al., 2015), Greenland Blocking Index (GBI; Hanna et al., 2016) and sea ice cover (Kopec et al., 2016). Wong et al. (2015) show that the first and second EOFs are associated with NAO variability, Hanna et al. (2016) link the first EOF to GBI, and Kopec et al. (2016) correlate the second EOF with sea ice cover. In addition, Greenland surface temperatures, which influence atmospheric moisture and precipitation processes, have been shown to be impacted by El Niño events (Gan et al., 2023; Matsumura et al., 2021).

To explore whether these large-scale climate drivers can be linked with the EOF patterns in this study, and to understand what their PC adjustment coefficients may signify, we compute correlations between the first 10 PCs and the following indices: NAO, based on 500-hPa geopotential height anomalies from the NOAA/CPC (Barnston and Livezey, 1987); GBI, defined as the mean 500-hPa geopotential height over 60°–80°N, 280°–340°E (Hanna et al., 2016); Northern Hemisphere sea ice area from the NSIDC Sea Ice Index v3.0 (Fetterer, F.; Knowles K.; Meier W.; Savoie M.; Windnagel A., 2017); and the Niño-3.4 index derived from ERSSTv5 SST anomalies over 5°N–5°S, 170°–120°W (Huang et al., 2017). Annual correlations are presented in Fig. 8 for the full periods of HIRHAM (1960-2022) and RACMO (1980-2022) output, as well as the 1991-2022 overlap period with CARRA (1991-2022).

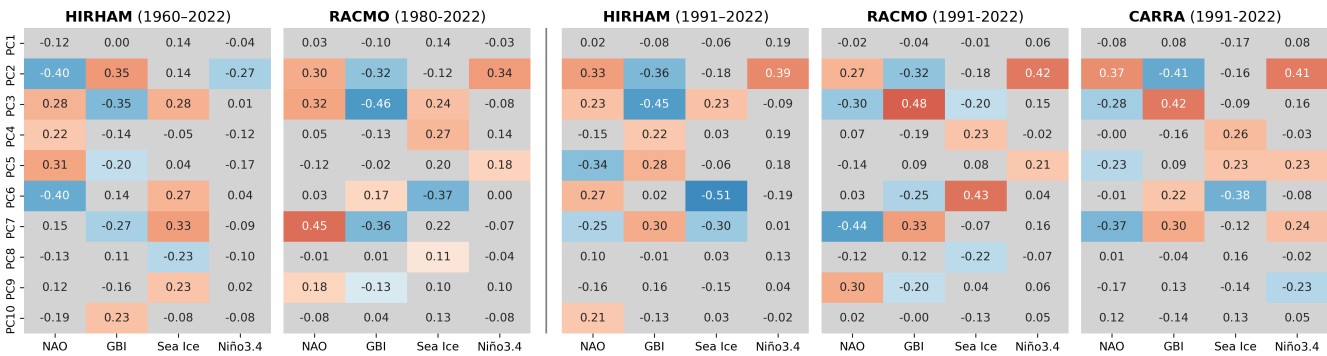

**Figure 8.** Annual correlations of the first 10 PCs with the NAO, GBI, Northern Hemisphere sea ice area, and Niño3.4 indices. Correlations are shown for the full HIRHAM period (1960–2022), RACMO period (1980–2022), and the overlap period with CARRA (1991-2022).

PC2 shows consistent correlations with NAO and GBI of between 0.3 and 0.4 in magnitude across all models and time periods, aligning with relationships identified in previous studies. Weaker but persistent correlations with PC2 and sea ice are also seen, ranging from 0.1–0.2 in magnitude. While not previously emphasised in relation to PC2, moderate positive correlations with ENSO across all three models may suggest it has a secondary influence on this mode, possibly through its teleconnections with North Atlantic atmospheric conditions. As the 2nd PC shows correlations with these four climate indices of similar magnitude across models and the corresponding EOF maps (Fig. A3) exhibit similar spatial patterns, this suggests that this mode represents similar pattern of variability across models, influenced by an interplay of these climatic circulation patterns.





The first PC only shows weak inter-annual correlations here. Wong et al. (2015), report correlations between NAO and
precipitation at individual coastal stations, particularly in western Greenland, with the strongest relationships found in winter.
However, they also note that the sign of the NAO–precipitation correlation varies regionally, leading to opposing responses
around the ice sheet margin. In contrast, our analysis is performed over the full ice sheet at an annual scale, potentially averaging
out seasonally varying regional signals. This is consistent with the spatial structure of the first EOFs (Fig. A3), which are largely
negative across the accumulation zone in all three models, suggesting a dominant mode of coherent variability that does not
capture regional contrasts.

As the PC adjustment coefficients are subject to the regularisation parameters, they deviate less than the EOF coefficients,
with few notable exceptions: HIRHAM PC3, RACMO PC4, and CARRA PCs 4 and 6, which are adjusted to 0.5, 0.2, 0.3,
and 2 respectively. A common feature among these PCs is their significant correlation with sea ice extent, which may suggest
a misrepresentation of the influence of sea ice variability on accumulation. This is particularly relevant near the coast, where
sea ice variability is known to strongly affect snowfall. The two-fold adjustment in CARRA 6th PC, which remains largely
unchanged in HIRHAM and RACMO, may be linked to the opposing mean bias patterns seen in CARRA versus HIRHAM
and RACMO along the north coast (Fig. 4). A similar argument can be presented for CARRA PC2 and PC3 coefficients, which
are adjusted down to 0.8 and 0.7, as a response to GBI.

While these examples do not prove causality, they may suggest potential links between specific modes of model bias and
known large-scale atmospheric patterns. However, it is important to note that EOFs and PCs are statistical constructs whose
ordering, spatial patterns and behaviour are not necessarily consistent across models. Therefore, while correlations with known
climate indices can offer insight, physical interpretation of the adjustments should be viewed within the context that EOFs are
not inherently structured to align themselves with physical drivers.

In practical terms, the EOF-based framework enables us to use the PCs derived from the accumulation zone to extrapolate
EOFs for the full ice-sheet (eq. 7). The EOFs could be further extended beyond the spatial limits of observational coverage to
infer bias-correcte precipitation/evaporation fields for the surrounding ocean, for example. In addition to spatial extrapolation,
the PCs may also offer the potential for temporal extrapolation through their relations with correlated time series, enabling
reconstructions of past accumulation patterns prior to availability of the RCM data. This could be implemented using a similar
approach to that described in Box et al. (2013), which regress RACMO2 output between 1958-2010 with data from 86 ice
cores to reconstruct net snow accumulation rates between 1600-2009. In our case, the bias-adjusted PCs could be regressed
against correlated climate drivers, temperature series, air pressure or ice-cores, for example. This approach could possibly
be further extended to obtain future estimates, using projections of such correlated variables, though the reliability of future
extrapolations would strongly depend on the confidence in the projected regression variables, as well as the magnitude and
stability of the underlying correlations.

## 5.4   Limitations

The reliability of the bias-corrected accumulation fields, associated temporal trends, temperature sensitivity estimates and any
reconstructions based on temporal extrapolations ultimately hinge on one main factor – the quality and confidence in the



observational data. Though the 1.7 million data points in SUMup provide substantial statistical power compared to smaller-scale studies, observational uncertainties ultimately limit the accuracy of these final products. Heavily dominated by annual and
multi-annual means, the SUMup dataset lacks sufficient monthly observations in the accumulation zone matching the temporal resolution of the model data. This, in combination with the dating uncertainties inherent in radar, ice-cores and snow pit derived accumulation estimates, limits the reliability of bias-adjusted accumulation values on sub-annual scales. In addition, radar and ice cores estimates require density conversions, which vary between datasets and introduce further uncertainties. To improve the accuracy of bias corrections, especially at sub-annual timescales, there is a pressing need for more high-
resolution, ground-truth data such as direct snow-water-equivalent observations that are independent of density assumptions. High-temporal-resolution measurements from emerging technologies, such as those based on cosmic ray sensing (e.g. Howat et al., 2018), offer a promising means of providing continuous, density-independent accumulation data.

## 6   Conclusions and Outlook

We have devised a novel statistical-semi-empirical framework to quantify and correct spatial and temporal biases in gridded
model accumulation using any in-situ observational data. Our method is applied here using observational SMB data from the SUMup dataset, to bias-adjust monthly accumulation output from three high-resolution models over the Greenland Ice Sheet: HIRHAM5 (5.5 km, 1960-2022), RACMO2.41 (11 km, 1980-2022), and CARRA reanalysis (2.5 km, 1991-2022). Relative to SUMup observations, we find point-wise initial mean biases of -8.7% (HIRHAM), +0.5% (RACMO) and +10.9% (CARRA). After bias-correction, all models converge to near-zero mean bias (-0.1% to -0.2%), with RMSE reduced by 8-
18% and correlation with observations improving by 1–3%. The resulting bias-corrected mean annual accumulation over the ice sheet are estimated at 321 mm yr$^{-1}$ (HIRHAM, 1960-2022), 375 mm yr$^{-1}$ (RACMO, 1980-2022) and 384 mm yr$^{-1}$ (CARRA, 1991-2022). By providing spatially complete, bias-corrected accumulation fields, the method offers improved inputs for ice-sheet mass balance studies and other modelling efforts, such as the Ice Sheet Model Intercomparison Project (ISMIP).

Given the calculated annual mean accumulation over GrIS is approximately 500 Gt yr$^{-1}$ (A1) and the total ice volume of
GrIS is ~2.5 million Gt (van den Broeke et al., 2016), a mean bias of 10% equates to 50 Gt yr$^{-1}$, or an annual error of ~0.002% of total GrIS volume per year. Over a century long run, a 10% bias accumulates to 5,000 Gt, with the potential to alter sea-level rise projections by approximately 15 mm by 2100 (The IMBIE Team, 2020). Using the empirical relationship between SMB and temperature from AR5 (Intergovernmental Panel On Climate Change, 2014, Chapter 13 Supplementary Material), a 50 Gt yr$^{-1}$ SMB deviation corresponds to an equivalent warming bias of ~0.7°C. This is a substantial error, especially when viewed
against critical thresholds for Greenland ice-sheet stability—such as the ~1.6°C threshold identified by Robinson et al. (2012), beyond which GrIS is expected to undergo irreversible long-term melt. Such SMB biases, if uncorrected, could obscure or misrepresent the proximity to tipping points in climate projections.

SMB biases can also indirectly affect the ice dynamical response of ice-sheet models. Errors in SMB are absorbed into model calibration of ice dynamical parameters and thereby affect projections. As ice flow is driven by spatial gradients in SMB,
accurate spatial distribution pattern of accumulation are essential for simulating realistic ice flow, geometry, and evolution.



Reliable projections are thus critically dependent not only on the total SMB, but on capturing its spatial variability across the ice sheet.

We compare the original and bias-adjusted accumulation maps to understand spatial bias patterns, identifying significant discrepancies between models. HIRHAM tends to underestimate across the accumulation zone, while CARRA largely overestimates across this domain. RACMO exhibits the lowest mean bias, but exhibits strong spatial contrasts, with a high positive bias in the south and negative bias further north. All three models show substantial bias in the south, consistent with previous studies, which have attributed substantial biases in the south to issues in resolving orographic precipitation over complex terrain (Langen et al., 2015; Box et al., 2013; Burgess et al., 2010).

These results underline the continued importance of bias-correcting model accumulation using in-situ observations to improve both the spatial fidelity and physical realism of SMB fields. However, the effectiveness of such corrections ultimately depends on the quality and resolution of the available observational data. The dominance of annual and multi-annual means and the scarcity of monthly-resolution data–ultimately limit the accuracy of the bias-adjusted accumulation at sub-annual timescales. To improve the accuracy of bias corrections, future efforts should prioritise expanding high-resolution in situ measurements, including direct snow-water-equivalent observations that are independent of density assumptions.

As our model is setup, new datasets can be readily integrated to further improve bias-correction. Here we have confined the bias-adjustment to the accumulation zone, enabling the use of consistent variables across the three models (precipitation minus evaporation/sublimation) to represent accumulation, rather than full SMB. This choice avoids reliance on uncertain snowpack or density modelling components, which are often inconsistent or unavailable. However, with sufficient observational constraints, the approach could be directly transferred to full SMB fields or other parametrisations and extended to the entire model domain. Our framework can also be adapted to provide bias-corrected fields for other climate variables, as well as other regions such as the Antarctic ice-sheet.



# Appendix A

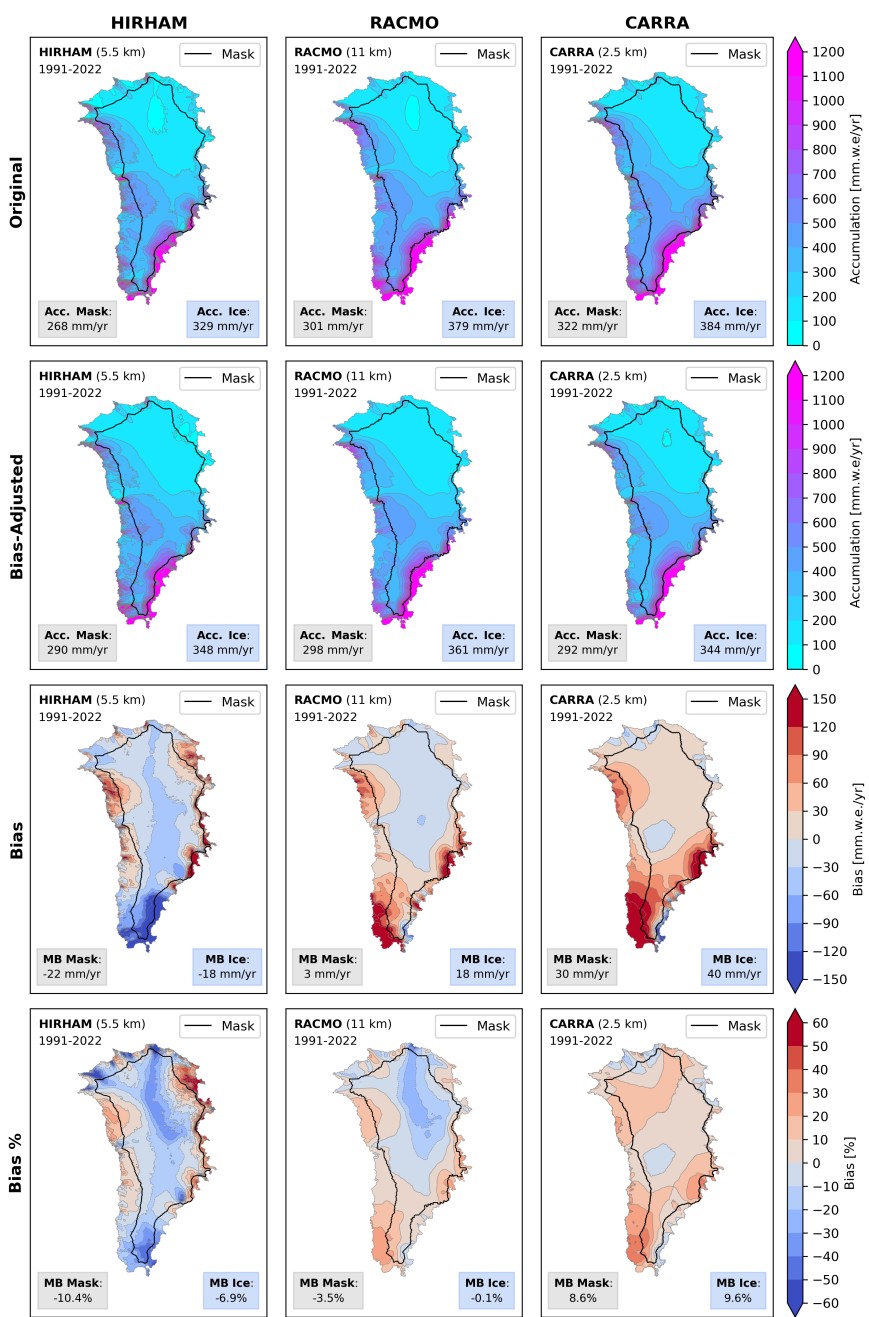

**Figure A1.** Mean maps of model accumulation and bias over the 1991-2022 overlap period. Values stated in the lower corners indicate the spatial mean over the accumulation zone (left) and ice sheet (right). First row: original mean accumulation before adjustment. Second row: mean bias-corrected accumulation. Third row: mean bias in mm yr$^{-1}$. Fourth row: mean percentage bias.



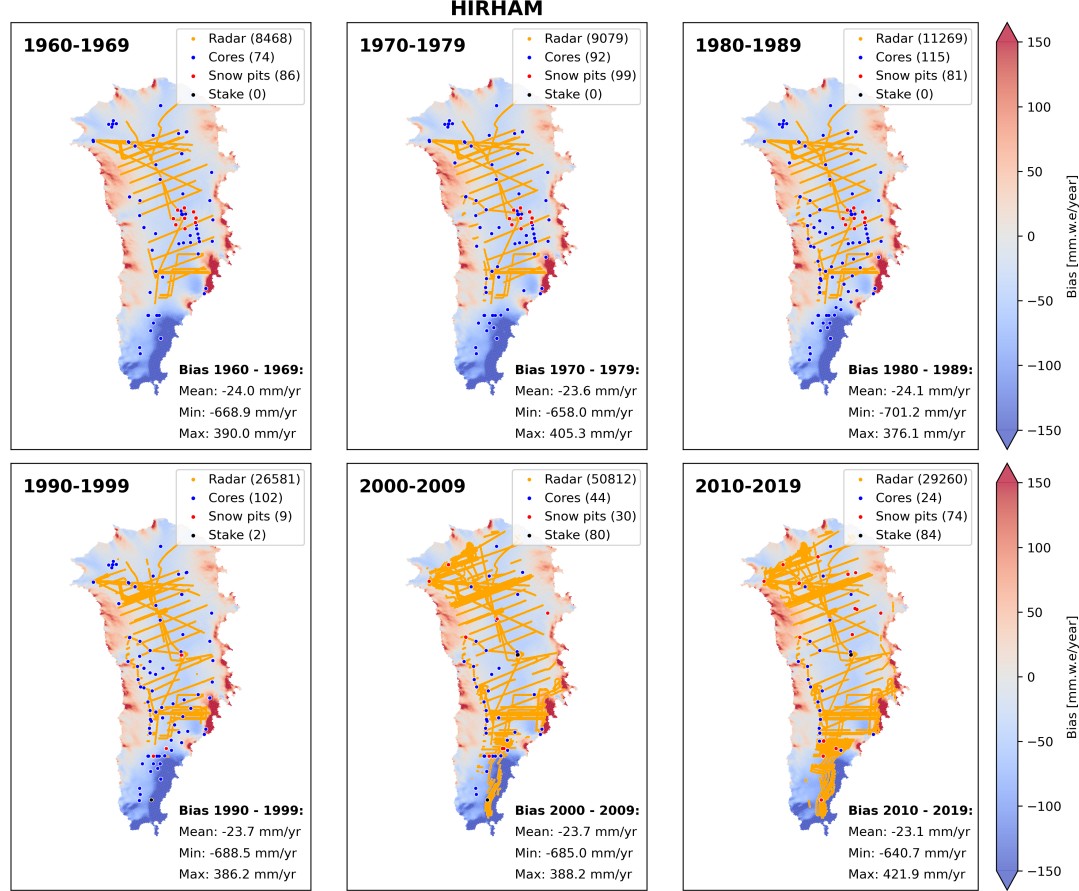

**Figure A2.** SUMup data distribution by decade, plotted over HIRHAM decadal bias with mean, minimum and maximum bias values printed for each decade.





**Figure A3.** First three EOF patterns for HIRHAM (1960-2022), RACMO (1980-2022) and CARRA (1991-2022) over the accumulation zone.




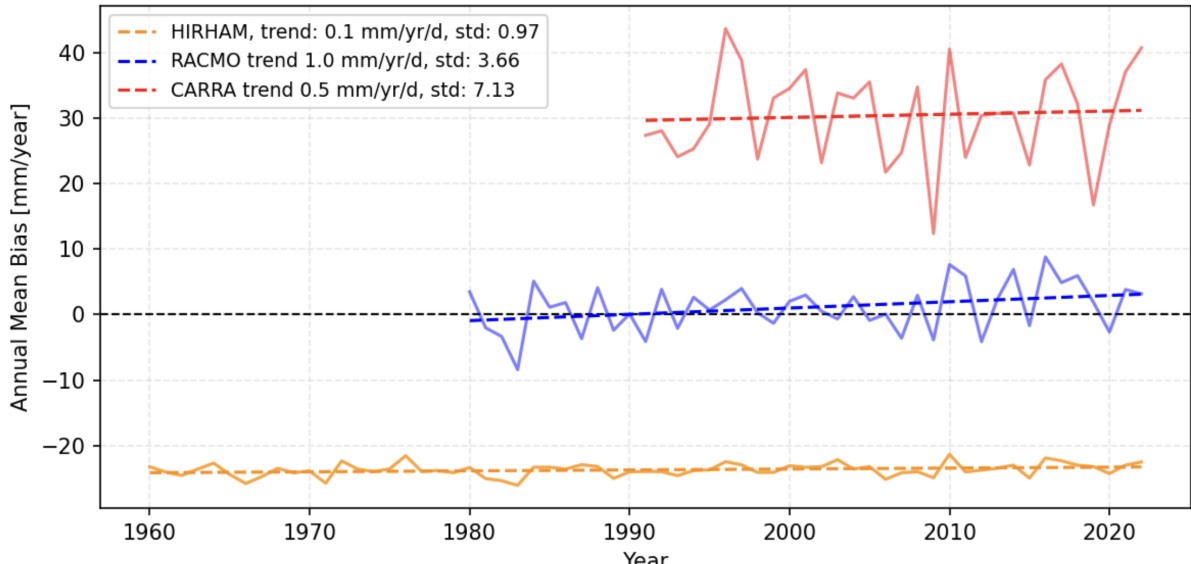

**Figure A4.** Mean annual bias for HIRHAM, RACMO and CARRA plotted against time with temporal trends and standard deviations.



**Figure A5.** Evaluation of model performance against observations from 6 ice core sites before and after bias-adjustment.





| | | Accumulation Zone | | | Ice Sheet | | |
| --- | --- | --- | --- | --- | --- | --- | --- |
| **Period** | **Model** | **Acc** $[\text{Gt yr}^{-1}]$ | **Acc$_{adj}$** $[\text{Gt yr}^{-1}]$ | **Bias** $[\text{Gt yr}^{-1}]$ | **Acc** $[\text{Gt yr}^{-1}]$ | **Acc$_{adj}$** $[\text{Gt yr}^{-1}]$ | **Bias** $[\text{Gt yr}^{-1}]$ |
| 1960–2022 | HIRHAM | 259 | 282 | −23 | 474 | 503 | −29 |
| 1980–2022 | HIRHAM | 263 | 287 | −24 | 481 | 511 | −29 |
| | RACMO | 301 | 300 | +1 | 560 | 536 | +24 |
| 1991–2022 | HIRHAM | 266 | 288 | −22 | 486 | 514 | −27 |
| | RACMO | 305 | 302 | +3 | 567 | 540 | +27 |
| | CARRA | 315 | 285 | +30 | 562 | 503 | +59 |

**Table A1.** Mean annual accumulation over the accumulation zone and ice sheet before and after bias-adjustment, including absolute bias, in Gt yr$^{-1}$





*Author contributions.* JLC and AG designed the bias-adjustment method. BV provided the figure of model evaluation against the 6 individual ice core sites. JLC wrote the initial manuscript, prepared and filtered the SUMup dataset, and created the remaining figures and tables. All
authors participated in the data interpretation and commented on the paper.

*Competing interests.* No competing interests are present.

*Acknowledgements.* JLC, AG and CH received funding from the Novo Nordisk Foundation (NNF Challenge, PRECISE – Prediction of Ice Sheets on Earth, grant no. NNF23OC0081251). CH received funding from the European Union (Horizon Europe, ICELINK, grant no. 101184621). The authors thank Martin Olesen (Danish Meteorological Institute) for providing HIRHAM5 data (1960-2022), Kristiina Verro
(Danish Meteorological Institute) for providing RACMO data (1980-2022), and Clément Cherblanc for valuable feedback on the manuscript.



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
