# Peer review of "Greenland Monthly Accumulation Maps (1960-2022): A Statistical Semi-Empirical Bias-Adjustment Model"

_EGUsphere, 2025_

## Author Comment (AC2)

Responses to Reviewers' Comments for Manuscript
EGUSPHERE-2025-2516

**Greenland Monthly Accumulation Maps (1960-2022):
A Statistical Semi-Empirical Bias-Adjustment Model**

Addressed Comments for Publication to

The Cryosphere

by

Josephine Lindsey-Clark, Aslak Grinsted, Baptiste Vandecrux, and
Christine Schøtt Hvidberg

The study presents a statistical model reconstruction for bias-corrected gridded regional model (e.g., HIRHAM and RACMO) and reanalysis model (CARRA) accumulation fields for the Greenland Ice Sheet (GrIS). The authors use an empirical orthogonal function (EOF) approach to reduce the model accumulation fields into primary spatial modes. Thereafter, the means, climatologies, EOFs and corresponding principal components (PCs) are adjusted through a set of coefficients that attempt to minimize the differences between modeled accumulation and SUMup in situ observations across the GrIS. Prior to adjustment, RACMO is shown to have the smallest absolute monthly mean accumulation biases, followed by HIRHAM and CARRA, but all accumulation zone and ice sheet-wide maps' biases are reduced, especially in RACMO, after the adjustment takes place. It is interesting to note that post-adjustment, the biases of the highest resolution product, CARRA at 2.5 km, are not reduced further. The authors' posit this may be due to such limited observations within these small gridbox areas.

The paper mainly showcases the statistical techniques and insights the render more accurate annual and seasonal accumulation maps and trends versus those produced with the native, unadjusted regional model/reanalysis outputs. This work may represent a significant advance toward understanding and developing Arctic/Antarctic accumulation maps going forward.

 The paper is nicely written, and results are by and large clearly presented. Several of my comments, however, are along the lines of structural changes where disentangling the tandem pairing of methods with or immediately followed by results is warranted. **A more clear outline of the paper's sections toward the end of the Introduction, followed by distinct separation of methods from results would improve the flow and readability of the manuscript.** Further comments are noted by line (L) number below. In summary, these collectively constitute at least minor revision.

**AC:** Thank you for your helpful comments and interest in our paper.

We agree the manuscript would benefit from a clearer outline of the paper's structure and a better separation of methods from results.

We will revise the end of the introduction to provide a clearer outline of the paper's structure, and address the distinction of methods and results as suggested in your specific comments.

**Major (Technical suggestions)**

**General comment:** While each regional model is shown for their respective dataset start years to 2022, what about also showing maps for consistent periods (i.e., the CARRA record), 1991-2022 for more direct comparison of seasonal patterns and trends? This could complement Table 3.

**AC:** To reduce the number of figures in the main text, the maps for 1991-2022 were provided in the appendix – '*Spatial patterns are shown here for HIRHAM (1960-2022), RACMO (1980-2022) and CARRA (1991-2022), and periods of overlap are provided in the appendix, Fig. A1'*. However, we agree the overlap period should be highlighted more clearly.

In the revised manuscript we will carefully consider whether to show the 1991-2022 overlap in the main text and provide the full model periods in the appendix instead.

**Minor (Typographical/Structural suggestions)**

**L83:** leavingàleave
**AC:** corrected

**L95-104:** This content seems like more methods and initial results than introductory material, particularly in the description of the number of principal components retained and their explained variance.

**AC:** We agree this interrupts the flow of the introduction and such details should be reserved for the methods section.

In the revised manuscript we will condense this and relocate more detailed information about the number of PCs and their explained variance to the methods section.

**L138:** What is meant by "is merged with" – please clarify.

**AC:** Thank you for the suggestion, we agree this is ambiguous.

We will clarify the wording to make the relationship between the variables more explicit in the revised manuscript.

**L156-166:** This comparison of RACMO P-E and SMB against SUMup precedes introduction of the SUMup data. These results should be given after SUMup is described in detail in 2.3 or as initial results within the Results section.

**AC:** Thank you for your comment. In response to Reviewer 2's feedback regarding snow erosion being substantial in parts of the ice-sheet, we will use SMB minus runoff rather than P-E, and so this discussion will be removed.

In the revised manuscript this discussion will be removed.

**L189:** some time à sometimes
**AC:** Corrected

**L193:** subject-verb agreement here needs corrected to the "pre-summer/post-summer end dates are"
**AC:** Corrected

**L328:** Why use HadCRUT5 versus another global temperature dataset such as Berkeley Earth or GISTEMP? A brief note justifying use and acknowledging shortcomings of this product is warranted here.

**AC:** Thank you for this comment. The three NH temperature reconstructions are very similar in the period considered in this study (post-1960), and we do not believe this choice has any impact on our conclusions.

We will perform a test to verify that our conclusions are robust to this choice. We will also include a brief justification for our choice of HadCRUT5 in the revised manuscript.

**L523:** overestimatesàoverestimate
**AC:** Corrected

**L623-630:** Seems like these large-scale climatic dataset descriptions should go in the data section, then a results subsection could be framed around climate relationships to the PCs.

**AC:** We agree that the descriptions of the large-scale climate datasets are more appropriately placed in the data section. However, to reduce the length of the manuscript, we will consider cutting or moving the discussion of the PC correlations to supplementary material in the revised version.

We will consider whether to remove or relocate the analysis of the PC correlations to supplementary material. If retained in the main text, we will introduce the large-scale climatic datasets in the data section and move the PC correlation table (Figure 8) to the results section after Figure 3 (bias-adjustment coefficients), with corresponding revisions to the text.

**L649-650:** "This is particularly relevant near the coast, where sea ice variability is known to strongly affect snowfall." A citation to previous work is needed here at the conclusion of the sentence.

**AC:** Thank you for the suggestion. We agree that adding relevant citations will strengthen this statement. These could include Shahi et al. (2023), who highlight how changes in sea ice impact local climate and SMB through temperature and snowfall effects in Northeast Greenland, and Kopec et al. (2016) who present observational evidence for the response of precipitation to sea ice reduction in the Arctic. However, to reduce the length of the manuscript, we will consider cutting or moving the discussion of the PC correlations to supplementary material in the revised version.

We will consider whether to remove or relocate the analysis of the PC correlations to supplementary material. If retained in the main text, we will add relevant citations to support this statement regarding the influence of sea-ice variability on coastal snowfall.

---

## Author Comment (AC3)

Responses to Reviewers' Comments for Manuscript
EGUSPHERE-2025-2516

**Greenland Monthly Accumulation Maps (1960-2022):
A Statistical Semi-Empirical Bias-Adjustment Model**

Addressed Comments for Publication to

The Cryosphere

by

Josephine Lindsey-Clark, Aslak Grinsted, Baptiste Vandecrux, and
Christine Schøtt Hvidberg

This study presents a statistical approach for bias-correcting accumulation data provided by regional climate models and atmospheric reanalysis. To do this, the authors first use an Empirical Orthogonal Function (EOF) to identify dominant modes of accumulation variation across the Greenland Ice Sheet. They then use the SUMup dataset to find a set of coefficients for adjusting the principal components (PCs) so that accumulation can be reconstructed without bias. Adjusting the PCs of accumulation grids allows the authors to derive accumulation biases across a much larger spatial extent and temporal resolution than would be possible using just the in-situ observations themselves. The manuscript is therefore able to identify model biases across the entire ice sheet in places that observations do not exist. Overall, I think that this is a very clever approach for adjusting modeled accumulation and the maps may represent our best current estimates of accumulation over the Greenland Ice Sheet.

I have a few general recommendations that I think would improve the manuscript.

**AC:** Thank you for your comments and interest in our paper. We appreciate your detailed suggestions, which will greatly improve the clarity and robustness of the manuscript. We have carefully addressed each comment as follows.

The first is that **the paper is quite long** which makes it challenging for the reader to follow at times. There are many repetitive statements that could be removed (or woven into previous text). There are also many cases where the authors provide figure (or table) "commentary" to start a section. **These commentaries (e.g. L223-224, L360-362, L371-376, L435-438, L462-466) interrupt the narrative of the results section and, in most cases, could easily be removed.** If the figures are clearly labelled and captioned, they don't need to be described again in the main text. I highlight the relevant lines in my specific comments below.

**AC:** Thank you for these suggestions of where to cut down. We agree the paper is quite long, and such commentary should be removed. In addition, it has become evident that we need to focus the discussion and remove/reorganise some parts of the analysis and discussion.

We will revise the manuscript to improve structure and readability, removing redundant figure/table commentary. Where appropriate, we will also move secondary material to the supplementary information and ensure that the main text maintains a clear, focused narrative.

My second recommendation is for **the authors engage a little more with the limitations of their analysis**. Currently **the limitations section is very short** (L6761-682) and only really discusses the uncertainties of the observational data. But there are other limitations that the authors mention earlier in the text which are not fully discussed in this section. I would be interested to see a

reflection about the use of P-E vs. SMB in this analysis. **The authors nicely show that P-E is perhaps better than SMB on average (Table 1). But there could be places on the ice sheet with substantial snow erosion or deposition where this may not be true.**

**AC:** We agree that although P–E appears better than SMB on average, it can be problematic in regions with substantial snow erosion or deposition. In response, we will instead use SMB (excluding runoff) throughout the analysis. Preliminary analysis shows that the change from P-E to SMB (excluding runoff) will not have a great impact on the spatial bias patterns, possibly because wind erosion was treated as a bias before, though it should in theory provide a more physically realistic representation of accumulation overall.

We will replace the use of P-E with SMB (excluding runoff) for RACMO and MAR, which have dedicated SMB variables. A description of how each model parametrises SMB will be added to the data section. The limitations section will be expanded, and include a more detailed discussion of the uncertainties mentioned throughout the manuscript.

Likewise, much of the **analysis hinges on the assumption that the bias-adjustments applied inside the study area (where SUMup data is available) can be applied to outside the study area (where SUMup data is absent).** I would like to see a **deeper discussion of the uncertainties involved** when **extrapolating beyond the measurements in flat, low accumulation areas to more topographically complex environments at the margins of the ice sheet.**

**AC:** Thank you for highlighting this important point. We acknowledge that extrapolating bias adjustment from the interior to topographically complex margins introduces additional uncertainty, particularly where accumulation regimes differ substantially. To address this, we will extend the bias-adjustment to the full ice sheet, using all relevant SUMup observations within a common ice-sheet mask (PROMICE-2022 Ice Mask; Lüetzenburg et al., 2025 – https://doi.org/10.22008/FK2/O8CLRE).

To maintain the focus on constraining accumulation and not runoff (for which the SUMup data are not sufficient), we will filter out negative SMB observations below an elevation of 2000 m, thereby excluding measurements likely resulting from melt/runoff. The filtering removes only negative measurements from in/near in the ablation zone, including observations from some stake measurements, SnowFox sensors, automated weather stations and those estimated from mass balance profile (i.e. Oerter et al., 1995a; Jung-Rothenhäusler et al., 1995; Bøggild et al., 1995; Podlech et al., 2004). The remaining negative SMB observations include only 5 stake measurements from Dibb and Fahnestock 2004 at Summit, which may result from snow erosion or compaction. Any

measurements whose quality are substantially impacted by melt are already excluded in the SUMup release. Therefore, we gauge that the remaining observations (Fig. 1) represent viable data for comparison with model output.

[Figure]

**Fig. 1:** Geographical distribution of relevant SUMup data within the PROMICE Ice mask (Luetzenburg et al, 2025). The accumulation zone is defined as the area within the mask defined in Vandecrux et al. (2019). The category 'other' includes the method types 'estimated from mass balance profile' and 'snowfox'.

We believe these changes will improve the accuracy and value of the resulting accumulation maps over the full ice-sheet, without implementing any changes to the method.

We will extend the bias-adjustment to the full ice sheet using all relevant SUMup observations, so that model output is bias-adjusted against conditions across the ice-sheet. The limitations section will be expanded accordingly to reflect the uncertainties associated with the observations and method.

Finally, I think the manuscript would be more readable if the authors stuck more rigidly to the IMRAD structure. **Some of the methods are in the results, results are in the discussion, and discussion points are in the conclusion**. I understand that there can be good reasons for doing this. At the same time, I've generally found that sticking to the conventional structure almost always results in a more concise and readable manuscript. I've highlighted most of these examples in my specific comments.

**AC:** We agree that the readability of the sections would benefit from restructuring to follow the IMRAD structure more closely, which would also aid in making the manuscript more concise.

We will restructure the manuscript as suggested, addressing your specific comments below.

**Specific comments**

**L4:** Do RCMs really contribute to "metre-scale" uncertainties in sea-level rise projections? In L697 the authors state that a 10% bias in snowfall could alter SLR projections by 15 mm.

**AC:** We appreciate this wording could be misleading.

In the revised manuscript, we will rephrase the statement to clarify that RCMs carry systematic temporal and spatially variable biases, which contribute to substantial uncertainties in sea-level rise projections.

**L5-6:** I'm don't think that any study has used reanalysis datasets to evaluate RCMs. Or are the authors saying that reanalysis datasets are considered to be in-situ? Either way, consider clarifying this sentence.

**AC:** Thank you for highlighting this. In response to reviewer 3's comments, we will reduce the amount of background material in the abstract and focus more directly on summarising key results. As such, this sentence will be removed.

We will remove this sentence from the abstract and merge the relevant parts of the background information from the first paragraph into the introduction.

**L10:** Describing the extent of SUMup using the number of points is misleading since many points are radar transects. Recommend describing SUMup in terms of temporal and spatial coverage or saying it is the most comprehensive dataset currently available.

**AC:** We agree that the strength of SUMup can better be described by its extensive coverage rather than the number of points.

We will revise this to reflect that SUMup represents the most extensive spatial and temporal coverage of GrIS surface mass balance observations to date.

**L11-12:** Applying an "EOF decomposition" to "adjust…EOFs" doesn't make sense, consider revising.

**AC:** We agree that this wording is unclear.

We will clarify this explanation of how the EOFs and PCs are used and adjusted based on SUMup in the revised manuscript.

**L16-17:** This is not really a major finding or very surprising. Consider replacing with a more impactful finding.

**AC:** We agree that the current abstract does not sufficiently highlight the most impactful findings. We will update the abstract to emphasise key results, such as quantitative changes in accumulation before and after bias adjustment, improvements in inter-model agreement, updated estimates of accumulation trends and their temperature sensitivity, as well as other principal findings emerging from the revised analysis.

We will revise the abstract to present the main quantitative and impactful findings of the study.

**L18-19:** These numbers appear to be incorrect according to Table 4. These look like the original accumulation rates.

**AC:** Thank you for highlighting this inconsistency.

These values will be updated to reflect the bias-adjusted accumulation rates consistent with the revised analysis.

**L25-26:** Slightly strange choice of references for this statement. I don't think any of them compared Greenland Ice Sheet mass loss against other sources so how can they show that it is the greatest? Thermal expansion is also technically the greatest single contributor so might consider using "cryospheric contributor" to acknowledge that point.

**AC**: We agree that Fettweis et al. (2020) and Hofer et al. (2020), which focus on projections rather than attribution, are not the best choice to support this statement. We will add Chen et al. (2017) and van den Broeke et al. (2017), both of which quantify contemporary sea-level-rise contributions and show that the GrIS is currently the largest cryospheric contributor. We will retain van den Broeke et al. (2016) – though it doesn't directly compare to other sources, this references GrIS as a major source. This paper is also used in Ryan 2020 to support a similar statement: '*The Greenland Ice Sheet has been losing mass at an accelerating rate since the start of the 21st century and is now the single largest cryospheric contributor to global sea-level rise (Chen et al., 2017; van den Broeke et al., 2016)*'.

We will revise the choice of references for this statement and rephrase the sentence to specify that the Greenland Ice Sheet is the largest *cryospheric* contributor to sea-level rise.

**L30-31:** Recommend including a reference for this sentence.

**AC:** Thank you for this suggestion, we agree this statement would be strengthened by a supporting reference.

We will add appropriate citations to support this sentence.

**L31-32:** I would argue that the "complexity" of precipitation patterns themselves does not make them hard to constrain. Instead, it is the "complexity of processes that cause precipitation" that are challenging to for models to constrain (related to grid cell resolution and simplified cloud microphysics).
**L32:** Poorly constrained "by models"? If so, I recommend clarifying.

**AC:** Thank you for this helpful clarification.

We will clarify that it is the complexity of the processes driving precipitation, along with model limitations such as resolution and parameterisations, that make accumulation over the GrIS challenging to constrain in models.

**L33:** The Hanna et al. (2024) is a review paper. I would prefer to see a citation to a study that has investigated this bias more directly (e.g. Ryan et al. 2020).

**AC:** Thank you for the suggestion.

We will replace Hanna et al. (2024) with Ryan et al. (2020).

**L47:** It's not immediately clear what the difference is between "accumulation patterns" and "spatial variability". Consider removing "spatial variability".

**AC:** Thank you for the suggestion, we agree using both is unnecessary.

We will remove the term 'spatial variability'.

**L58:** Does climate reanalysis assimilate in-situ atmospheric observations over Greenland? Recommend including a citation to provide evidence for this statement.

**AC:** We agree that this statement is ambiguous. Our intention was to convey that, unlike RCMs, reanalyses assimilate a variety of observations globally and in the arctic. In some cases this includes Greenland, for example, Hanna et al. (2005) note that DMI surface air temperatures are assimilated into the surface scheme used to produce the ECMWF reanalysis, though there is no direct reference to support this statement in that paper.

We will clarify this in the revised manuscript.

**L60:** "Limitations in model physics" is pretty vague, can the authors be more specific about causes of snowfall bias in models? Is it grid cell resolution failing to capture complex, steep topography? Is it simplified cloud microphysics?

**AC:** Thank you for this suggestion.

We will revise this to specify that biases arise from factors such as coarse grid resolution limiting the representation of complex, steep terrain and simplified cloud microphysics parameterisations.

**L68-69**: "better representation" by the model or by the ice cores?

**AC:** Thank you for this comment.

We will revise this to clarify the improvement is due to the hybrid approach outlined in Burgess et al. (2010).

**L77:** SMB is different to accumulation which is what the previous paragraphs focused on. Consider replacing "SMB" with "accumulation".

**AC:** Thank you for this suggestion.

We will change 'SMB' to 'accumulation'.

**L81-82:** I don't think it's fair to completely dismiss remote sensing technology but then cite a study that has measured snowfall using remote sensing. It can be done with CloudSat, it's just challenging because of sampling limitations and ground clutter (see Ryan et al., 2020).

**AC:** We agree the current text sounds as if we are dismissing remote sensing technology, which was not our intention.

We will revise the text to acknowledge that remote sensing can measure snowfall, though it faces challenges such as sampling limitations and ground clutter, including a citation to Ryan et al. (2020).

**L83:** Not sure the authors can say "today" and reference a paper from 2013. An update is provided by Ryan et al. (2020) and there may be a more recent study.

**AC:** We agree a citation to a more recent study is needed.

We will add Ryan et al. (2020) to support this statement.

**L85:** I would argue that the challenges with in-situ measurements are spatial more than anything (i.e. limited spatial coverage and uncertainties caused by point-to-pixel differences).

**AC:** Thank you for this suggestion.

We will revise this sentence to emphasise that uncertainties caused by point-to-pixel differences, combined with challenges in aligning the inconsistent temporal resolutions, means that the full range of available data remains under-utilised in systematic RCM validation.

**L88:** Would it be fair to say that this study advances previous studies by incorporating the SUMup dataset for bias correction? If so, I recommend that the authors make this point clearer in the introduction.

**AC:** Thank you for highlighting this.

We will adapt the end of this paragraph to reflect that our model advances prior work through leveraging the full range of in-situ observations contained within the SUMup data.

**L95-102:** This is all interesting but it interrupts the flow of the introduction, which at this stage should be setting up the aims of the paper. Consider moving this to the methods.

**AC:** We agree this interrupts the flow of the introduction and such details should be reserved for the methods section.

In the revised manuscript we will condense this section and relocate more detailed information about the number of PCs and their explained variance to the methods section.

**L107:** "accumulation" - do the authors mean "SMB" here?
**L107:** This sentence seems a little debatable given the challenges of accounting for runoff - recommend removing and putting in the discussion where it can be argued more convincingly.

**AC:** Thank you for noting this.

These sentences will be removed as our revised approach will fit the whole ice-sheet. We will explain that we will not bias-adjust runoff as this cannot be constrained with SUMup.

**L141:** I think this section needs some sort of explicit statement that says "we assume accumulation = SMB" in our study area.

**AC:** Thank you for this comment.

This section will be replaced in our updated manuscript in accordance with our revised approach.

**L141:** I also recommend stating the percentage of the ice sheet area and the amount of total ice sheet snowfall that falls in this region. My back-of-the-envelope calculation from MAR suggests the area where runoff is zero is 73% of the ice sheet and 58% of the precipitation.

**L150:** I think the term "accumulation zone mask" is a little misleading here because the real accumulation zone is actually much bigger than this (i.e. runoff is greater than zero but still less than snowfall). I recommend replacing the term with "study area" or something similar so as not to confuse a reader who only skims the paper.

**AC:** Thank you for these comments, which were an additional motivation for us to expand our study area to the whole ice-sheet.

We will instead delineate the accumulation zone as the area falling within the accumulation zone mask defined in Vandecrux et al. (2019) (https://doi.org/10.5194/tc-13-845-2019). We will provide statistics on the area and snowfall percentages for the accumulation zone and ablation zone.

**L163**: "assess"
**AC:** Corrected

**L171:** Is this the actual accumulation zone or the "accumulation zone mask" that was defined previously?

**AC:** This referred to the accumulation zone mask, which will not be used in our revised approach.

We will remove this sentence in the revised manuscript.

**L187-190:** Are the authors implying that the radar data represent less than 12 months of accumulation? If so, they could be more explicit about that. Relatedly, do the authors use the radar data "as is" or is a correction applied to the radar data to account for the fact that the snow layer represents less than 12 months? It would be useful to clarify that here.

**AC:** We agree our original wording could be misinterpreted. We did not mean that we apply a correction, rather that we shift the start/end dates from July to September, meaning that deeper radar layers still represent 12 months. For surface layers, the end dates correspond to the date the measurement was taken and thus represents accumulation from the previous horizon in September, until the measurement date. We rely on the dates as tabulated in SUMup, but to allow for uncertainty in dating the horizons, we assign a dating uncertainty of ±12 months, as described in lines 209-215.

We will clarify this explanation in the revised manuscript.

**L95-102:** Detailed description about EOF technique should be moved to the methods.

**AC:** We agree this interrupts the flow of the introduction and such details should be reserved for the methods section.

In the revised manuscript we will condense this section and relocate more detailed information about the number of PCs and their explained variance to the methods section.

**Figure 2:** Should RACMO not be blue (not orange)?

**AC:** The histograms are layered such that HIRHAM (orange) is plotted behind RACMO (blue), which are then behind CARRA (red) – the histogram outline in blue denotes where the RACMO residuals lie. However, we agree this could be presented more clearly.

We will make the distinction between model residuals more clear in the updated figure and clarify how to interpret the histogram in the figure caption.

**L277-278:** the three "ands" in this sentence makes it difficult to understand what is being adjusted.

**AC:** Thank you for this comment, we agree this sentence doesn't read well.

We will rephrase this in the updated manuscript to clarify which parts of the decomposition are being adjusted by each coefficient.

**Figure 3:** The difference between the yellow vs. orange and dark blue vs. light blue is really not clear. Suggest separating on another panel or changing color scheme

**AC:** We agree the clarity of this figure could be improved.

We will present only the scaling coefficients for the 1991-2022 overlap period in this figure, and provide the coefficients for the full model periods in the appendix.

**L371-376:** See general comment but this is example of a paragraph that could be removed without losing much. The reader should be able to deduce all this information from the figure and caption alone.

**AC:** Thank you for your suggestions of where to cut down, we agree the manuscript should be condensed where possible and such paragraphs should be removed.

We will remove this paragraph and carefully consider other similar passages which could be cut down.

**L499:** Fig. 3?
**AC:** Corrected

**L512:** "overestimate"
**AC:** Corrected

**L498-513:** This is all description of the findings so should be placed in the Results section. The Discussion really starts at L513.

**AC:** We agree this descriptive section is better suited to the results section.

We will cut this section from the discussion and merge the necessary details with the results.

**L515:** In Greenland? Please clarify.

**AC:** Thank you for this comment.

We clarify that this citation studies Greenland.

**L527-528:** This is pretty vague. Could the authors be more specific about how higher spatial resolution enhances "sensitivity to intense precipitation events"? Consider adding citations as evidence.

**AC:** We agree this requires more explanation with a citation to support the statement.

We will clarify that models with higher spatial resolutions can better resolve topography and therefore orographic precipitation, and add citations to support this.

**L531:** Ryan et al. (2020) confirms this bias using CloudSat.

**AC:** Thank you for suggesting this reference.

We will add Ryan et al. (2020) as an additional citation.

**L536-537:** This is a confusing sentence because it seems to imply that the bias is larger in the southwest but it looks like the authors are only considering the area of bias. Recommend the authors quantify the total bias to investigate whether snowfall is more biased in the southeast or southwest (i.e. in km3 or Gt).

**AC:** Thank you for highlighting this ambiguity. We understand that referring only to the spatial extent of the bias could imply that the southwest has a larger bias overall, even if the magnitude differs between regions.

In the revised manuscript, we will clarify that we refer to the area of bias here, and also quantify the magnitude of bias in the southeast and southwest.

**L540:** Again, recommend the authors quantify the contribution of northern interior bias, perhaps as a percentage.

**AC:** We agree that stating that the northern interior contributes significantly to model bias should be supported by the value of the contribution.

In the revised manuscript, we will quantify the magnitude of the bias when referring to the contribution of a particular region.

**L549:** Repetition of previous paragraph, consider removing.

**AC:** We agree there is some repetition of previous information.

We will rephrase this section to avoid repetition.

**L551:** "are seen" is poor wording.

**AC:** Thank you for the comment.

We will rephrase this to improve the wording.

**L553-554:** This seems important because this bias should not be attributed to model. Can the authors provide a reference for this?

**AC:** Thank you for this comment.

We will add citations for the tendency for deeper ice-core layers to be more prone to miscounting (Steig et al. 2005), and radar horizons being more likely to be misidentified (Bingham et al. 2025).

**L556:** Please provide a reference for this statement. I understand that reanalysis assimilates observations but are there any observations that are assimilated over Greenland?

**AC:** We agree that this statement should be supported by a reference

We will add a citation to Herbach et al. (2019).

**L561:** This seems like pure speculation. Can the authors provide a reference as evidence for winder conditions in the winter and spring months?
**L564-565:** Why would more frequent storms make them more likely to be "lifted over steep terrain"?

**AC:** We agree that speculative statements such as these should be supported by relevant citations. To improve focus, we will restructure the discussion to emphasise key findings, and consequently, we will consider removing or substantially revising this section to reduce length and improve clarity.

We will restructure the discussion to focus on key results, only including more speculative interpretations if they can be supported with relevant citations.

**L565-567:** I'm not sure I follow, doesn't RACMO overestimate snowfall in the southeast? Surely a finer resolution model would overestimate snowfall more because the topography is steeper? Please consider citing a figure (or reference) as evidence for this statement.

**AC:** Thank you for highlighting this ambiguity.

We will carefully revisit this statement in the revised analysis and support it by quantifying the bias values, with specific references to the figure and a citation if appropriate.

**L573:** This is sentence is just a repetition of the previous sentence. Doesn't reduction in accumulation trends by bias-adjustment show that the models are overly sensitive to warming air temperatures?

**AC:** We agree that this sentence is repetitive and could instead be rephrased to reflect that the reduction in accumulation trends after bias-adjustment suggests that models may be overly strong sensitivity to warming air temperatures.

We will revisit this statement in the revised analysis, rephrasing to clarify the implications of the change in accumulation trends resulting from the bias-adjustment.

**L574:** "are seen" is poor wording.

**AC:** Thank you for the comment.

We will rephrase this to improve the wording.

**L582:** The more robust trends in HIRHAM could be simply caused by the longer time period. Recommend comparing models over common time period before dismissing trends in RACMO and CARRA.

**AC:** We agree this sounds as if we are dismissing these trends based on the model rather than stating the HIRHAM trends are simply more robust due to the longer time period.

We will revise this text to reflect that the HIRHAM trends are the more robust simply due to the longer time period, rather than due to model differences.

**L583-588:** I'm not sure that this paragraph adds much so it could be deleted.

**AC:** Thank you for the suggestion.

We will remove this paragraph from the revised manuscript.

**L603:** I would encourage the authors to reserve the word "significantly" for statistical statements.

**AC:** Thank you for highlighting this.

We will revise the wording to avoid using 'significantly' except when referring to statistical significance.

**L604:** "may point" is poor wording

**AC:** Thank you for the comment.

We will rephrase this to improve the wording.

**L616-653:** This text presents new findings so should be moved to the Results section.

**AC:** We agree that this material presents new findings and would be better placed in the results. However, to reduce the length of the manuscript, we will consider cutting or moving the discussion of the PC correlations to supplementary material in the revised version.

We will consider whether to remove or relocate the analysis of the PC correlations to supplementary material to reduce the length of the manuscript. If retained in the main text, we will relocate this content to the Results section.

**L619:** first and second EOFs "of snowfall"?

**AC:** Thank you for highlighting this.

We will clarify that we refer to EOFs of precipitation.

**L662:** Can the authors be more specific about what they mean by "correlated time series"? Air temperature? SST?

**AC:** We agree this is unclear. We will clarify that the PCs are limited to the temporal extent of the model output and therefore cannot be used for long-term reconstructions. Instead we can model the temporal variability associated with the PCs based on correlated proxies such as those explored in Fig. 8 (NAO, GBI, Sea Ice, ENSO), as well as other proxies such as air temperature, air pressure, SST, ice cores, lake sediments and tree rings.

We will revise the text to specify the types of correlated time series we are referring to and why they would enable reconstruction of PC variability beyond the model period.

**L691-692:** These look like the original numbers, not the bias-adjusted. It would also be better to use a common time period like in Table 4 so the comparison is "apples-to-apples".

**AC:** Thank you for the suggestion and for identifying this inconsistency.

We will update these numbers to match the bias-adjusted values and state the values for the 1991-2022 overlap period for direct comparison.

**L694-702:** This is all interesting but the Conclusion section shouldn't introduce new information. It would be better to place this text in the Discussion.

**AC:** We agree this should be introduced in the discussion.

We will introduce this content in the discussion and may refer back to it in the conclusion.

**References**

Luetzenburg, Gregor; Korsgaard, Niels J.; Deichmann, Anna K.; Socher, Tobias; Gleie, Karin; Scharffenberger, Thomas; Meyer, Rasmus P.; Fahrner, Dominik; Nielsen, Eva B.; How, Penelope; Bjørk, Anders A.; Kjeldsen, Kristian K.; Ahlstrøm, Andreas P.; Fausto, Robert S., 2025, "PROMICE-2022 Ice Mask", https://doi.org/10.22008/FK2/O8CLRE, GEUS Dataverse, V2

Ryan, J. C., Smith, L. C., Wu, M., et al. (2020). Evaluation of Cloudsat's cloud-profiling radar for mapping snowfall rates across the Greenland Ice Sheet. Journal of Geophysical Research: Atmospheres, (4), e2019JD031411.

Vandecrux, B., MacFerrin, M., Machguth, H., Colgan, W. T., van As, D., Heilig, A., Stevens, C. M., Charalampidis, C., Fausto, R. S., Morris, E. M., Mosley-Thompson, E., Koenig, L., Montgomery, L. N., Miège, C., Simonsen, S. B., Ingeman-Nielsen, T., and Box, J. E.: Firn data compilation reveals widespread decrease of firn air content in western Greenland, The Cryosphere, 13, 845–859, https://doi.org/10.5194/tc-13-845-2019, 2019.

---

## Author Comment (AC4)

Responses to Reviewers' Comments for Manuscript
EGUSPHERE-2025-2516

**Greenland Monthly Accumulation Maps (1960-2022): A Statistical Semi-Empirical Bias-Adjustment Model**

Addressed Comments for Publication to

**The Cryosphere**

by

Josephine Lindsey-Clark, Aslak Grinsted, Baptiste Vandecrux, and
Christine Schøtt Hvidberg

**General comments**

Lindsey-Clark et al. describe a statistical method to bias-correct and present arguably the most accurate estimates to date of mean annual and seasonal 1960 to 2020 Greenland Ice Sheet (GrIS) snow accumulation rates. The work is motivated by GrIS snow accumulation uncertainty, most of which the authors attribute to poor observation sampling and limited underlying model resolution in regional climate models (RCMs) and in reanalyses. The methodology, a principal component analysis (PCA) applied using GrIS accumulation estimates from RCMs, the Copernicus Arctic Regional Reanalysis (CARRA), and observations of SMB components, is compelling and well executed. The figures and tables are labeled and presented clearly and, combined with the conclusions, are of good quality.

Depending on its applicability to the next generation of GrIS precipitation hind-casts and as forcing data for dynamic ice sheet and climate models, the manuscript presents a good level of substantial progress and novelty and could be fairly significant to the community researching ice sheet-climate interactions. The PCA and resulting improvements to observation-based RMSE scores are well suited for GrIS snow accumulation dependent studies and should be interesting to readers of *The Cryosphere*. On the other hand, the authors discuss their own findings in detail but not in the context of a substantially relevant body of literature. A more developed discussion of more recent work would improve the manuscript's fair to poor scientific quality in this regard.

**AC:** Thank you for your comments and interest in our paper. We have carefully addressed each comment as follows.

**Specific comments:**

1. The abstract is heavy on background material and light on quantitative results. To improve conciseness, the authors could **cut most of lines 2-8**, most of which could be merged with the introduction or removed altogether. While the authors present over 40 subpanels of maps and plots plus several meaningful statistics throughout the body of the manuscript, few of these results are reported in the abstract. To improve completeness, the authors should consider **summarizing more of their results in the abstract or submit for publication with their manuscript a supplement that repurposes extraneous figures**. I suggest that the authors consider which subset of their numerous results are essential to the main findings (i.e., those reported in the abstract and conclusions) and **streamline the presentation of data by moving figures that can be interpreted more as supporting evidence to a supplement.**

**AC:** We agree that the background material in the abstract could be condensed, and selected details merged with the introduction. We also agree that the manuscript is quite long and would benefit

from a more focussed presentation and discussion of the main results. We will carefully consider which figures to retain in the manuscript, and move selected figures and discussion, such as the PC correlations analysis, to supplementary material.

We will restructure the manuscript and aim for a more compact presentation of the most important results and findings, moving more exploratory analyses, such as the PC correlations, to supplementary material. We will adapt the abstract to reflect the more compact selection of results, stating explicitly the study's main findings while condensing the background information. The conclusions will be revised to reiterate the main results highlighted in the abstract. Additionally, we will streamline the text by removing unnecessary figure/table commentary to avoid repetition and improve the clarity of the manuscript.

2. To improve the overall presentation quality, parts of the **data section (section 2) could benefit from better organization**, which suffers from a **poor balance of details regarding the gridded data** sets compared with the observations. **Much of section 2.3 could be cut and merged with material in section 5**, which would improve the discussion of other recent related work and better enable readers' understanding of the broader context.

**AC:** We agree that the description of the gridded datasets would benefit from additional detail. We intend to keep the presentation of SUMup (section 2.3) in the data section, but we will consider whether specific sections, such as the pre-bias-adjustement histograms of model-SUMup residuals, would be better placed in the results section or moved to supplementary material.

We will expand the descriptions of model datasets to provide necessary additional detail. We will carefully evaluate which parts of section 2.3 could be either cut down, relocated to the results section, or moved to supplementary material.

3. In my view, the work's significance and its application to GrIS precipitation estimates depends on it being reproducible and/or the accessibility of the resulting data sets for use with impact studies (e.g., forcing data for firn, ice sheet, and sea level rise modeling and projections). To improve the work's potential impact, I recommend that the authors **clarify details of their input data and methods where possible.** For example, there are **lacking details regarding the RCMs and CARRA data sets (section 2.1)** and of **the applied regularization process (lines 291-296). HIRHAM is mentioned several times but no citation appears until section 2.1.1**. **Regarding the RCM and CARRA data sets, there are so few details provided that an unfamiliar reader must refer to other sources to learn what the acronyms represent** and to understand basic components of the modeling systems, let alone their more technical nuances and strengths and weaknesses regarding

estimating GrIS precipitation rates. While I understand that the authors need not fully describe previously published data sets, the level of details regarding these "main characters" is severely out of balance compared to that provided for the observational data (section 2.3). And while the detail of the methodological description is sufficient, due to the size of data sets and complexity of algorithms involved, full traceability of results would almost certainly require that the **authors submit for publication with their manuscript archived source codes and/or software containers** that have been tested on commonly used computing platforms and operating systems.

**AC:** We agree that the potential impact and reproducibility of the study would be improved by providing clearer descriptions of the input datasets and methodological steps. While we are unsure whether we can redistribute raw model output, we will review the relevant terms to confirm what is permissible. Either way, we will provide a minimal working code package that can be run using, for example, the SUMup data including our applied dating uncertainties, as well as the EOFs and PCs derived from the raw model data.

We will expand the descriptions of model datasets to provide necessary additional detail, clarifying all acronyms at first use and adding an earlier citation for HIHRAM. Additionally, we will revise the description of the regularisation procedure to include additional detail such as the grouping used in cross-validation and the training/validation functions. In the description of the regularisation process, we will include additional details such as the grouping used for the cross-validation and the training and validation functions. We will supply a minimal working code to enable users to repeat the analysis with relevant input data.

4. The authors' discussion of other work **lacks consideration of underlying physical processes**, **quantification of modeling errors and uncertainty**, and **limitations of data assimilation methods used by reanalysis systems**. Beyond their own findings, discussion of mean annual and seasonality of biases and trend and EOF analysis includes only four studies published within the last seven years **(van der Schot et al., 2024; Box et al., 2023; Gan et al., 2023, Matsumura et al., 2021). Underlying data assimilation challenges inherited by reanalyses, sources of model biases inherited by both reanalyses and RCMs, and North Atlantic teleconnections are closely related areas of research but are not discussed in the manuscript.** For example, the authors mention that **GrIS surface temperature variability and its related impact on precipitation is driven by El Niño events but do not elaborate on the "...teleconnections with North Atlantic atmospheric conditions…." (line numbers 620-635) nor develop the current body of research in this area.** To improve scientific quality, I suggest the authors **refine the discussion of their results with respect to the context of more recent relevant work,** particularly regarding **driving processes**, **uncertainty quantification and speculations of error sources in fit-for-purpose models and reanalyses** (including those considered and not considered in their study).

**AC:** We agree that a more complete treatment of uncertainty would strengthen the scientific quality of the manuscript. As the uncertainty covariance of the SUMup dataset is unknown, we will investigate approaches to quantify uncertainty in our results. Specifically, we may use the spread in the cross-validation fits for the optimum regularisation parameter to derive uncertainty estimates for key reported statistics, including those highlighted in the abstract. However, for some derived metrics, it may be more appropriate to characterise uncertainty using the spread across the four bias-adjusted model outputs, and we will consider this where relevant.

We agree that speculative statements regarding driving processes or teleconnections should be supported by appropriate citations. To improve clarity and focus, we will restructure the discussion to emphasise the key findings of the study. As part of this restructuring, we will consider removing the section on PC correlations and teleconnections (including lines 620-635) along with other discussion on driving processes, which are exploratory and not essential to the main objectives of the paper.

The primary scope of this paper is to present our methodology and provide improved, data-informed accumulation estimates. A detailed investigation of the physical mechanisms driving biases, data assimilation limitations, or North Atlantic teleconnections–though closely related areas–is beyond the intended scope. Nevertheless, where we retain results that touch on related topics, we will ensure that they are properly contextualised with reference to the relevant recent literature, such as van der Schot et al. (2024), Box et al. (2023), Gan et al. (2023) and Matsumura et al. (2021).

We will introduce uncertainty estimates for key statistics. We will restructure the discussion to focus on the core findings and carefully consider whether to remove discussion of the PC correlations, teleconnections, and other speculative interpretations of driving processes, or place them in supplementary material. We will revise any remaining discussion of physical drivers to ensure it is supported by appropriate recent literature and consistent with the scope of the study.

**Technical corrections:**

**1. Line 15:** Please define the acronyms HIRHAM5, RACMO, and CARRA in the abstract.

**AC:** Thank you for this comment.

In the revised manuscript we will define the acronyms for all models in the abstract.

**2. Line 29:** Further articulate the SMB definition(s), considering if the terms "specific SMB" and/or "climatic mass balance" are more appropriate for use with this manuscript.

**AC:** Thank you for suggesting this clarification. We will review the use of 'SMB', 'specific SMB' and 'climatic mass balance' to ensure that our definitions are consistent with glaciological conventions and with the characteristics of the data used in this study.

We will consider carefully that our definitions are correct and sufficiently specific for the introduction and whether further clarification in the observation–model comparison sections is needed to express how these definitions relate to the model output and observational records used in this study.

**3. Lines 103-105**: The purpose of the work, which is to improve GrIS SMB estimates and downstream modeling of sea level rise, is not stated accurately or is misrepresented.

**AC:** Thank you for highlighting this. We agree that this key aim of the study is currently not highlighted in the introduction.

In the revised manuscript, we will ensure that this section of the introduction clearly situates the method and bias-corrected accumulation fields within the context of improving GrIS SMB estimates and downstream modelling of sea-level rise.